**Regime-based Aerosol-Cloud Interactions from CALIPSO-MODIS and the Energy Exascale**
**Earth System Model version 2 (E3SMv2) over the Eastern North Atlantic**
Xiaojian Zheng[1], Yan Feng[1], David Painemal[2], Meng Zhang[3,a], Shaocheng Xie[3], Zhujun Li[2,4], Robert
Jacob[1] and Bethany Lusch[5]
[1]Environmental Science Division, Argonne National Laboratory, Lemont, IL, USA
[2]Science Directorate, NASA Langley Research Center, Hampton, VA, USA
[3]Lawrence Livermore National Laboratory, Livermore, CA, USA
[4]Analytical Mechanics Associates, Hampton, VA, USA.
[5]Argonne Leadership Computing Facility, Argonne National Laboratory, Lemont, IL, USA
[a]Now at: Department of Earth and Atmospheric Science, University of Houston, Houston, TX, USA
**Correspondence**: Xiaojian Zheng (zhengx@anl.gov)
**Abstract.** This study investigates aerosol-cloud interactions in marine boundary layer (MBL) clouds
using an advanced deep-learning-driven synoptic-regime-based framework, combining satellite data
(CALIPSO vertically resolved aerosol extinction and MODIS cloud properties) with 1° nudged Energy
Exascale Earth System Model version 2 (E3SMv2) simulation over the Eastern North Atlantic (ENA;
~10°×10°, 2006-2014). The E3SMv2 captures observed seasonal variations in cloud droplet number
concentrations ($N_d$) and liquid water path (LWP), though it systematically underestimates $N_d$. We then
partition ENA meteorology into four synoptic regimes (Pre-Trough, Post-Trough, Ridge, Trough) via a
deep-learning clustering of ERA5 reanalysis fields, enabling regime-dependent aerosol-cloud
interactions analyses. Both satellite and E3SMv2 exhibit an inverted-V LWP–$N_d$ relationship. In Post-
Trough and Ridge regimes, the satellite shows stronger negative LWP–$N_d$ sensitivities than in Pre-Trough
regime. The Trough regime displays a muted satellite LWP response. In comparison, the model predicts
more exaggerated LWP responses across regimes, with LWP increasing too quickly at low $N_d$ and
decreasing more sharply at high $N_d$, especially in Pre-Trough and Trough regimes. These exaggerated
model LWP sensitivities may stem from uncertainties in representing drizzle processes, entrainment, and
turbulent mixing. As for $N_d$ susceptibility to aerosols, $N_d$ increases with MBL aerosol extinction in both
datasets, but the simulated aerosol-cloud interactions appear oversensitive to meteorological conditions.
Overall, E3SMv2 better captures aerosol effects under regimes that favor stratiform clouds (Post-Trough,
Ridge), but performance deteriorates for regimes with deeper, dynamically complex clouds (Trough),
highlighting the need for improved representations of those cloud processes in climate models.

## 1. Introduction

Marine boundary layer (MBL) clouds play a pivotal role in regulating the Earth's energy budget due to their extensive coverage over the oceans and high albedo (Albrecht et al., 1995; Wood et al., 2015; Dong et al., 2023; Wall et al., 2023). Central to the quantification of the radiative impact of MBL cloud properties is to determine the sensitivity of clouds to the presence of aerosols. Aerosols can impact cloud microphysical properties, such as the cloud droplet number concentration ($N_d$) and droplet effective radius ($r_e$), and, consequently, alter cloud optical and macrophysical properties, including liquid water path (LWP) and cloud fraction (Twomey, 1977; Albrecht et al., 1989; Zheng et al., 2020; Dedrick et al., 2025). The interactions between aerosols and clouds, commonly termed aerosol-cloud interactions (ACI), contribute to one of the largest uncertainties in climate projections (IPCC, 2021). These uncertainties in the aerosol-induced cloud microphysical responses and the accompanying cloud adjustments stem from the inherent complexity of cloud microphysical processes (e.g., droplet activation, precipitation suppression, and entrainment-induced evaporation) and their interactions with dynamically evolving MBL, where aerosol perturbations can potentially contribute to either the brightening or darkening of clouds (Wall et al., 2022; Feingold et al., 2024).

Satellite remote sensing observations are essential in efforts to quantify the cloud adjustment to aerosol perturbations, by providing spatially extensive datasets. Numerous studies using satellite data have demonstrated a significant relationship and progressively advanced our understanding of cloud adjustments to aerosols (Bellouin et al., 2020; Diamond et al., 2020; Yuan et al., 2023; Feingold, et al., 2025; Goren et al., 2025). Observational evidence frequently shows that, at lower $N_d$ conditions, with the increased aerosol concentration, cloud droplets become more numerous in smaller sizes. It leads to decrease in efficiency in colliding and coalescing into raindrops and suppress precipitation (Albrecht, 1989). This suppression results in less cloud water being lost through rainfall and, consequently, an increase in LWP. Furthermore, the combined effects of entrainment-sedimentation feedback and precipitation-stabilization may lead to weaker entrainment drying in relatively clean clouds, further enhancing the LWP under lower $N_d$ conditions (Bretherton et al., 2007; Wood, 2012; Possner et al., 2020). In contrast, at higher $N_d$ levels, the increased abundance of small droplets expands the surface area available for evaporation at the cloud top (Gupta et al., 2021; Zhang et al., 2022; Zheng et al., 2022a). This enhancement in evaporation promotes cooling and intensifies localized turbulent mixing, which, in turn, facilitates the entrainment of dry air from above the cloud. The subsequent mixing further accelerates the evaporation of cloud droplets, reducing the overall liquid water content and decreasing

LWP. However, distinguishing causality from correlations is a persistent challenge, as atmospheric variability, retrieval biases, and sampling limitations can obscure true aerosol-induced effects (Arola et al., 2022; Goren et al., 2023; Liu et al., 2024).

Parallel to observational advances, the modeling community has made significant progress in simulating aerosol-cloud interactions within global climate models (GCMs). Recent GCM versions incorporate more physically based cloud microphysics parameterizations, which enable the simulation of precipitation suppression and enhanced entrainment responding to aerosol changes. For instance, studies by Mülmenstädt et al. (2024a) show that some of the GCMs in the Coupled Model Intercomparison Project Phase 6 (CMIP6), namely the US Department of Energy (DOE) Exascale Earth System Model version 1 (E3SMv1; Golaz et al., 2019) and others, can capture the inverted V-shaped relationship between LWP and $N_d$, as often observed from satellite retrievals, although discrepancies persist regarding the causal interpretation of these relationships. Tang et al. (2024) highlights that even when E3SM version 2 (E3SMv2; Golaz et al., 2022) simulates the overall cloud macrostructure, the microphysical responses to aerosol perturbations are still subject to systematic uncertainties related to precipitation processes and turbulence–microphysics interactions. Collectively, these studies illustrate both the strides made and challenges remained in representing aerosol-cloud interactions in large-scale models. While some GCMs may successfully replicate observed negative LWP-$N_d$ relationships under the present-day conditions, they struggle to accurately simulate the turbulence, entrainment, and precipitation feedback, that govern cloud adjustments to aerosol changes (Mülmenstädt et al., 2024b). Assessing model performance in simulating cloud microphysical responses to aerosol perturbations using observation is a crucial step toward improving the process-level understanding.

Accurate assessment of aerosol effects on cloud microphysics faces several hurdles. Methodological inconsistencies in sampling, aerosol proxies, and comparison metrics limit direct observation-model comparisons. Studies that use vertical aerosol extinction profiles rather than column integrated aerosol optical depth report stronger links between aerosols and cloud microphysics, highlighting the need for refined observational strategies (Painemal et al. 2019, 2020). Furthermore, satellite retrieval errors, vertical mismatch between aerosol and cloud layer, updraft variability, and precipitation effects, can bias the estimates of cloud sensitivity to aerosol perturbations (Quaas et al., 2020; Gryspeerdt et al., 2022; Jia et al., 2022; Alexandri et al., 2024). Uncertainties in cloud adjustment processes, particularly the balance between precipitation suppression and entrainment driven evaporation, remain a persistent source of discrepancy between observational inferences and model simulations (Mülmenstädt et al. 2024b, Zhang and Feingold 2023). These challenges are compounded by the complex, multiscale nature of ACIs, where small-scale processes interact nonlinearly with larger-scale

meteorological drivers. Moreover, synoptic systems organize boundary-layer clouds on multi-day
timescales and strongly modulate aerosol-cloud-precipitation pathways (Mechem et al., 2018; Lee et al.,
2025). Therefore, quantifying the untangled aerosol-cloud sensitivities require conditioning on the
synoptic environment. For example, Zhang et al. (2022) found that the relationship between LWP and
$N_d$ is not only sensitive to aerosol loading but also modulated by the underlying meteorological conditions.
And McCoy et al. (2020) used a cyclone compositing approach to demonstrate that aerosol-cloud
interactions (e.g., the sign of LWP change with $N_d$) can differ inside vs. outside midlatitude cyclones.
These considerations motivate our use of an objective synoptic-regime classification to control
meteorology when evaluating the synoptic-regime-dependent ACI.
The Eastern North Atlantic (ENA) region is uniquely advantageous for advancing our
understanding of ACIs in MBL clouds (Wood et al., 2015; Tian et al., 2025). Located at the confluence
of subtropical and midlatitude air masses, the ENA is characterized by diverse meteorological conditions
and cloud regimes. This region frequently experiences well-organized stratocumulus cloud decks and
other MBL cloud types, which are sensitive to both local and long-range transported aerosols (Wang et
al., 2020; Wang et al., 2022). Observations document distinct aerosol and cloud properties, and the
relatively pristine marine background punctuated by episodic aerosol events helps separate aerosol driven
cloud adjustments from meteorological variability (Zheng et al., 2022b; Varble et al. 2023; Christensen
et al. 2024; Qiu et al., 2024). Moreover, the ENA has been extensively sampled, with long-term
observational data collected at multiple spatiotemporal resolutions from various platforms including the
DOE's Atmospheric Radiation Measurement (ARM) research facility (Wood et al., 2015), and several
satellite remote sensing products such as those from the Cloud Aerosol Lidar and Infrared Pathfinder
Satellite Observations (CALIPSO; Winker et al., 2009, 2010), and the MOderate Resolution Imaging
Spectroradiometer (MODIS) on board Terra and Aqua  (Barnes et al., 1998; Platnick et al., 2003; King
et al., 2013). These comprehensive observational datasets make the ENA an ideal testbed for evaluating
model aerosol-cloud interactions.
In this study, we employ a novel regime-based evaluation framework that combines the recently
developed CALIPSO-derived vertical-resolved aerosol extinction profiles and MODIS-derived cloud
properties with simulations from the DOE E3SM version 2 (E3SMv2). By applying a clustered-
meteorology-regime-based analysis over the ENA, we aim to isolate the impact of aerosol perturbations
on cloud microphysical properties by regime and seek to provide a robust observational and modeling
framework for quantitatively assessing aerosol-induced cloud adjustments. This approach is not only
intended for reconciling discrepancies between satellite observations and model simulations but also for
informing about potential model uncertainties in ACIs in connection with specific meteorological
regimes. The data and methods used in this study are introduced in Section 2. The seasonality of aerosols,
clouds, and their interactions are introduced in Section 3. And more importantly, Section 4 presents the
meteorological-regime-based analysis of the aerosol-cloud interaction and cloud adjustments. Section 5
summarizes conclusions with discussion, and outlines future work.
**2. Data and Method**
**2.1 Satellite retrievals of aerosols and clouds**
In this study, aerosol extinction coefficient ($\sigma_{EXT}$) is a research product derived from the Cloud-
Aerosol Lidar with Orthogonal Polarization (CALIOP) on the CALIPSO. These retrievals are produced
at a 1 km along-track resolution using the Fernald–Klett iterative approach, constrained by an
independent CALIOP-based aerosol optical depth (AOD), and are limited to cloud-free pixels. The
retrieved profiles have been evaluated against airborne High Spectral Resolution Lidar (HSRL)
measurements during the Caribbean 2010 field campaign and show good agreement. The detailed aerosol
retrieval methodology and product evaluation are described in Painemal et al. (2019) and Li et al. (2022).
Cloud properties, including the liquid water path (LWP) and the cloud droplet number
concentration ($N_d$), are obtained from MODIS Aqua at 1 km resolution using Clouds and Earth's Radiant
Energy System (CERES) Edition 4.0 algorithms (Minnis et al., 2021). The MODIS-retrieved LWP is
estimated to have uncertainties of approximately 10–15% when compared with ARM surface-based
observations (Xi et al., 2014; Painemal et al., 2016). The $N_d$ is retrieved using the adiabatic formulation
(Painemal and Zuidema, 2011; Grosvenor et al., 2018):
$$N_d = \Gamma^{1/2} \frac{10}{4\pi \rho_W^{1/2} k} \frac{\tau^{1/2}}{r_e^{5/2}} \; ,$$
Where the cloud droplet effective radius ($r_e$) and cloud optical depth ($\tau$) are estimated from
MODIS 3.79 μm and 0.64 μm bands, respectively (Painemal et al., 2013, 2020). $\Gamma$ denotes the adiabatic
lapse rate of condensation (Albrecht et al., 1990), which is calculated from the cloud-top temperature
and pressure derived from MODIS (Painemal et al., 2020). $\rho_W$ is the water density and $k$ represents the
ratio between the cloud droplet volume mean radius and the effective radius, assumed to be constant at
0.8 (Martin et al., 1994).
Although the relative errors in $N_d$ retrieval can be significant at the pixel scale (Grosvenor et al.,
2018), previous studies have shown that the $N_d$ compares well with measurements from 11 aircraft
campaigns, demonstrating a decent correlation when sampling the marine stratocumulus clouds, with $r^2$
values of 0.5~0.8 (Gryspeerdt et al., 2022). Therefore, to minimize known retrieval uncertainties, we
focus on low-level liquid clouds where satellite $N_d$ shows the strongest aircraft agreement and typical
normalized root mean squared deviation of ~30-50 % (Gryspeerdt et al., 2022). Moreover, the aggregated
collocation method significantly reduces the MODIS Aqua $N_d$ bias (Painemal et al., 2020), resulting in a
relationship between aerosol and cloud properties less affected by artifacts. Note that to avoid diurnal
variations in aerosol-cloud relationships, we fix the sampling to the Aqua local-afternoon overpass and
do not merge with Terra morning orbits, while extending the collocation and quality-control framework
to Terra is left for future work.
To collocate aerosol and cloud retrievals from the two satellite datasets, the following matching
method is employed. For each 1-km CALIOP aerosol pixel, five 1-km MODIS pixels are selected on
each side of the CALIPSO track (thus 10 MODIS pixels in total). The CALIOP retrievals are first
averaged to produce a 5 km along-track resolution product, and the MODIS cloud retrievals are
aggregated into four 5 km × 5 km grids (two grids east and two west of the CALIPSO track). These
datasets are then further averaged over approximately 25 km along-track segments, ensuring that the
aerosol and cloud data are matched at similar spatial scales. Hence, the clear-sky aerosol extinction
profiles can be collocated in the vicinity of the clouds, enabling the 'simultaneous' assessment of aerosol-
cloud interaction from the satellite data around 1 p.m. local time over ENA region. In addition, the
CloudSat Cloud Profiling Radar (CPR) is used to determine precipitation status from the satellite. The
drizzle condition is defined as the ratio of pixels with maximum radar reflectivity between –15 and –7
dBZ to the total number of pixels (19) within the 25 km satellite collocated segment, while the thresholds
for light rain and rain conditions are defined as maximum radar reflectivity greater than –7 dBZ but less
than zero dBZ, and greater than 0 dBZ, respectively. For a detailed description of the data-matching
strategy, please refer to Painemal et al. (2020) and Li et al. (2025).
To simplify terminology, all aerosol and cloud properties retrieved from the different satellite
products are hereafter referred to collectively as "satellite" data.

**2.2 E3SM simulations**

E3SMv2 is a fully coupled Earth System model (Golaz et al., 2022). Its atmospheric component, EAMv2, closely follows its predecessor EAMv1, as described in Rasch et al. (2019) and Xie et al. (2018), with only minor updates to its physical parameterizations. EAMv2 employs a spectral element dynamical core with approximately 110 km horizontal resolution and 72 vertical layers. The radiation and aerosol treatments in EAMv2 follow the Rapid Radiative Transfer Model (RRTM; Mlawer et al., 1997) and the four-mode version of the Modal Aerosol Module (MAM4; Liu et al., 2016; H. Wang et al. 2020), respectively. Turbulence, shallow convection, and cloud macrophysics are handled by the Cloud Layers Unified by Binormals (CLUBB) scheme (Larson, 2017; Golaz et al., 2022), while stratiform cloud microphysics is simulated by the Morrison-Gettelman (MG2) scheme (Gettelman and Morrison, 2015). Deep convection is represented by the Zhang and McFarlane Scheme (Zhang and McFarlane, 1995), as in EAMv1, but with a revised convective triggering function in EAMv2 that improves the simulation of precipitation and its diurnal cycle (Xie et al. 2019).

In this study, EAMv2 was run at standard resolution (~110 km) with the meteorology nudged to ERA5. The model was nudged toward the ERA5 zonal (U) and meridional (V) wind and temperature fields using a relaxation time of 6 h. This nudging reduces errors in the simulated large-scale state, facilitating the examination of aerosols and clouds. The nudged simulations reduce errors in simulated meteorological conditions, facilitating the examination of aerosol and clouds. Hourly outputs are available from 2006 to 2014 over the ~10°×10° ENA domain (33.5–43.5°N, 23–33°W), comprising 54 model columns. The $\sigma_{EXT}$ profile is directly outputted from the model. The MBL cloud samples are defined below 680 hPa to better match the satellite observations, and a cloud fraction threshold greater than 5% is used to determine a valid MBL cloud layer as in Kang et al. (2024). To further compare with the MODIS-retrieved cloud-top height (CTH), CTH is inferred by the diagnosed inversion height in E3SM. The inversion height is determined where $\left(\frac{\partial \theta_l}{\partial z}\right)\left(\frac{\partial RH}{\partial z}\right)$ is minimized, with the $\theta_l$ denoting liquid-water potential temperature and $RH$ denoting relative humidity derived from the model outputs. Given the coarse vertical resolution of E3SM near the cloud top (~200-300 m), this approach accounts for strong thermodynamic inversions and the effects of entraining dry air from the free troposphere (Erfani et al., 2022). The cloud-base height (CBH) is similarly identified using the 5% cloud fraction threshold. The in-cloud $N_d$ is obtained from grid-box-averaged cloud liquid number, weighted by cloud fraction at each

vertical level. Lastly, the cloud LWP is computed by integrating the in-cloud liquid water content (LWC)
between cloud-top (CTH) and cloud-base (CBH) levels: $LWP = \int_{CBH}^{CTH} LWC \, dz$.

**2.3 Clustering Model**

Clustering meteorological patterns allows us to systematically characterize and categorize the

diverse atmospheric conditions that modulate aerosol-cloud interactions in the ENA region. By
identifying distinct meteorological regimes, we can, to some extent, isolate the aerosol-driven
microphysical responses from the meteorological variability. In this study, we adapted an advanced deep
learning-based clustering model proposed by Faruque et al. (2023), which features the architecture of the
convolutional neural networks (CNN), and long short-term memory (LSTM) layers combined with a
Deep Embedded Clustering (DEC) framework. This hybrid CNN-LSTM-DEC model was designed to
capture complex spatiotemporal dependencies in meteorological data, overcoming limitations associated
with conventional clustering methods that often treat spatial and temporal features separately (Zheng et
al., 2025).

The clustering model was applied to ERA5 reanalysis data over a ~10°×10° domain (33.5–43.5°N,

23–33°W) in the ENA region for 2006–2014, with data at 13:00 LT (1 p.m. local time) daily to better
match the time of satellite records. Input variables included 500 hPa geopotential height (Z500), mean
sea level pressure (SLP), and the 10-meter u and v wind components. The CNN-LSTM-DEC architecture
employs a sequence-to-sequence autoencoder with an encoder comprising four convolutional blocks
(with filter sizes of 64, 128, 256, and 512), each followed by max-pooling to distill spatial features. An
LSTM layer with 512 units captures temporal dependencies, and a dense layer with 256 units (using
ReLU activation) defines the compressed latent space. A key aspect of our study was the fine-tuning of
model hyperparameters through the grid search technique, which enabled us to systematically optimize
clustering performance. The optimal configuration utilized a Stochastic Gradient Descent (SGD)
optimizer with a learning rate of 0.01, momentum of 0.95, and a batch size of 32.

We first pretrained the CNN-LSTM autoencoder using a reconstruction loss. Then, the latent

features extracted by the encoder were clustered using K-means, with which clustering into four groups
yielded the highest silhouette score of 0.267, compared to scores of 0.257, 0.178, and 0.167 for five, six,
and seven clusters, respectively. Hence, the determination of final cluster numbers (four, in this study) is
based on a combination of silhouette score analysis (i.e., measures of cluster cohesion and separation for
different cluster numbers) and the sensitivity of aerosol and cloud property distinctions to the chosen
number of clusters.
To further refine the clustering assignments, we then ran DEC with that fixed cluster number of
four, as determined with K-means optimization. DEC was initialized by the K-means centroids and
optimized the KL-divergence clustering loss (between soft assignments and a sharpened target
distribution) with periodic centroid updates, which increased the silhouette score to 0.358, indicating
enhanced cluster cohesion and separation. This two-step clustering process significantly reduced intra-
cluster variability while accentuating differences between clusters, as suggested by Faruque et al. (2023).
Their work also demonstrated that integrating both CNN and LSTM layers produces more robust latent
representations, which are the reduced-dimensional encoding of the input that captures its most
significant attributes, compared to CNN-only models or traditional approaches such as K-means and self-
organizing maps. Furthermore, Zheng et al. (2025) showed that including additional meteorological
variables, notably the 10-meter wind components, improved the model's ability to distinguish subtle
synoptic regimes over the ENA region compared to studies that considered Z500 only (e.g., Mechem et
al., 2018). Overall, the refined CNN-LSTM-DEC model demonstrates a marked improvement in
clustering performance over traditional methods for analyzing large-scale meteorological phenomena.

## 3. Aerosol and cloud properties from satellite and E3SMv2

### 3.1 Seasonal distribution of cloud properties

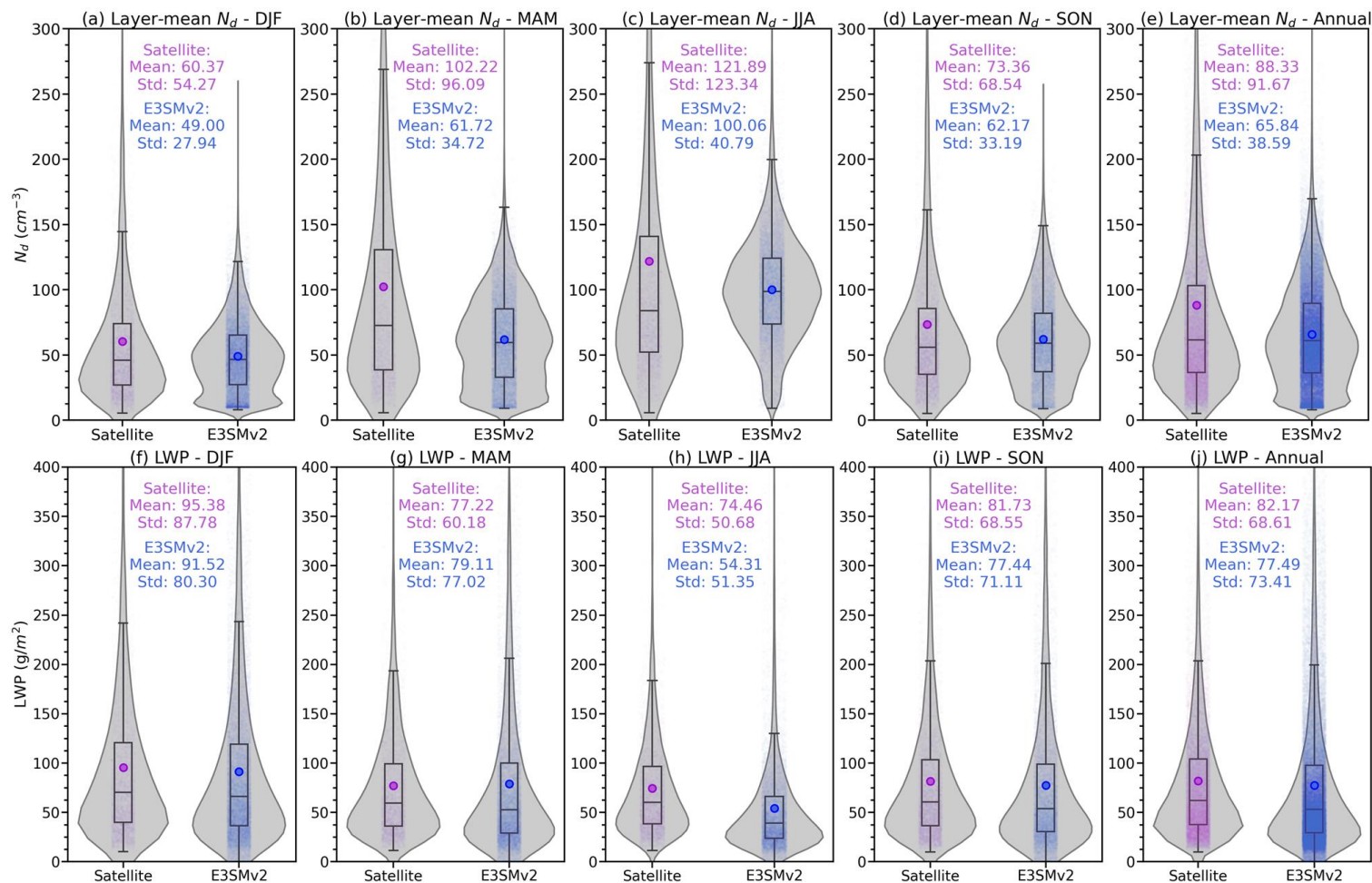

**Figure 1.** Violin plots of cloud droplet number concentration ($N_d$, top panels) and cloud liquid water path (LWP, bottom panels) from satellite retrievals (purple) and E3SMv2 simulations (blue), during winter, DJF (a, f), spring, MAM (b, g), summer, JJA (c, h), fall, SON (d, i), and Annual (e, j). The mean value is indicated by the color-coded dot. The smoothed shape of each violin shows the Gaussian kernel density estimate (KDE). From top to bottom within each violin, the box plot lines represent the third quartile (Q3, 75th percentile), median (Q2, 50th percentile), and first quartile (Q1, 25th percentile), respectively. The upper whisker extends to Q3 + 1.5 × IQR (interquartile range), and the lower whisker extends to Q1 − 1.5 × IQR.

Figure 1 illustrates the seasonal variations in the $N_d$ and LWP for low-level clouds over the ENA,
from satellite (MODIS) retrievals and E3SMv2. Annual means are $88.33 \pm 91.67$ cm$^{-3}$ and $82.17 \pm 68.61$
g m$^{-2}$ for satellite $N_d$ and LWP, and $65.84 \pm 38.59$ cm$^{-3}$ and $77.49 \pm 73.41$ g m$^{-2}$ for E3SMv2 (Fig. 1e
and 1j). Satellite-derived $N_d$ exhibits a pronounced annual cycle, with the highest mean values during
summer (JJA, 121.89 cm$^{-3}$) and spring (MAM, 102.22 cm$^{-3}$), followed by fall (SON, 73.36 cm$^{-3}$), and
the lowest during winter (DJF, 60.37 cm$^{-3}$). E3SMv2 reproduces the $N_d$ trend with a JJA peak 100.06
cm$^{-3}$ and a DJF minimum 49.00 cm$^{-3}$ but underestimates $N_d$ in every season (Fig. 1a-1d), with the largest
low biases in MAM 40.5 cm$^{-3}$ and JJA 21.83 cm$^{-3}$. Satellite observations display broader distributions
with higher variability especially during MAM (96.09 cm$^{-3}$) and JJA (123.34 cm$^{-3}$). In contrast, the
E3SMv2 exhibits generally narrower $N_d$ distributions, with distinct peaks at low concentrations in all
seasons except JJA. The model bias of overproducing frequent low $N_d$ scenarios in MBL clouds in
previous E3SM versions remains in E3SMv2 (Varble et al., 2023; Tang et al., 2023; Kang et al., 2024).
Satellite LWP in Fig. 1e to 1h varies modestly with a DJF maximum (95.38 g m$^{-2}$) and a JJA minimum
(74.46 g m$^{-2}$). Spread is greatest in DJF and smallest in JJA. E3SMv2 captures the phase with DJF (91.52
g m$^{-2}$) and JJA (54.31 g m$^{-2}$) and matches seasonal means best in DJF and SON, yet the distributions are
more positively skewed, indicating a tendency for more frequent low LWP.
Overall, higher Nd and lower LWP in warm seasons and the opposite in cold seasons agree with
prior ground based, satellite, and aircraft evidence (Wu et al., 2020; Varble et al., 2023; Zheng et al.,
2024). E3SMv2 successfully captures the broad seasonal variations in both $N_d$ and LWP, but
underestimates the amplitude and variability of $N_d$, especially during the warm seasons, pointing to
needed refinements in aerosol cloud interactions and microphysical process representation.

## 3.2 Responses of LWP to $N_d$

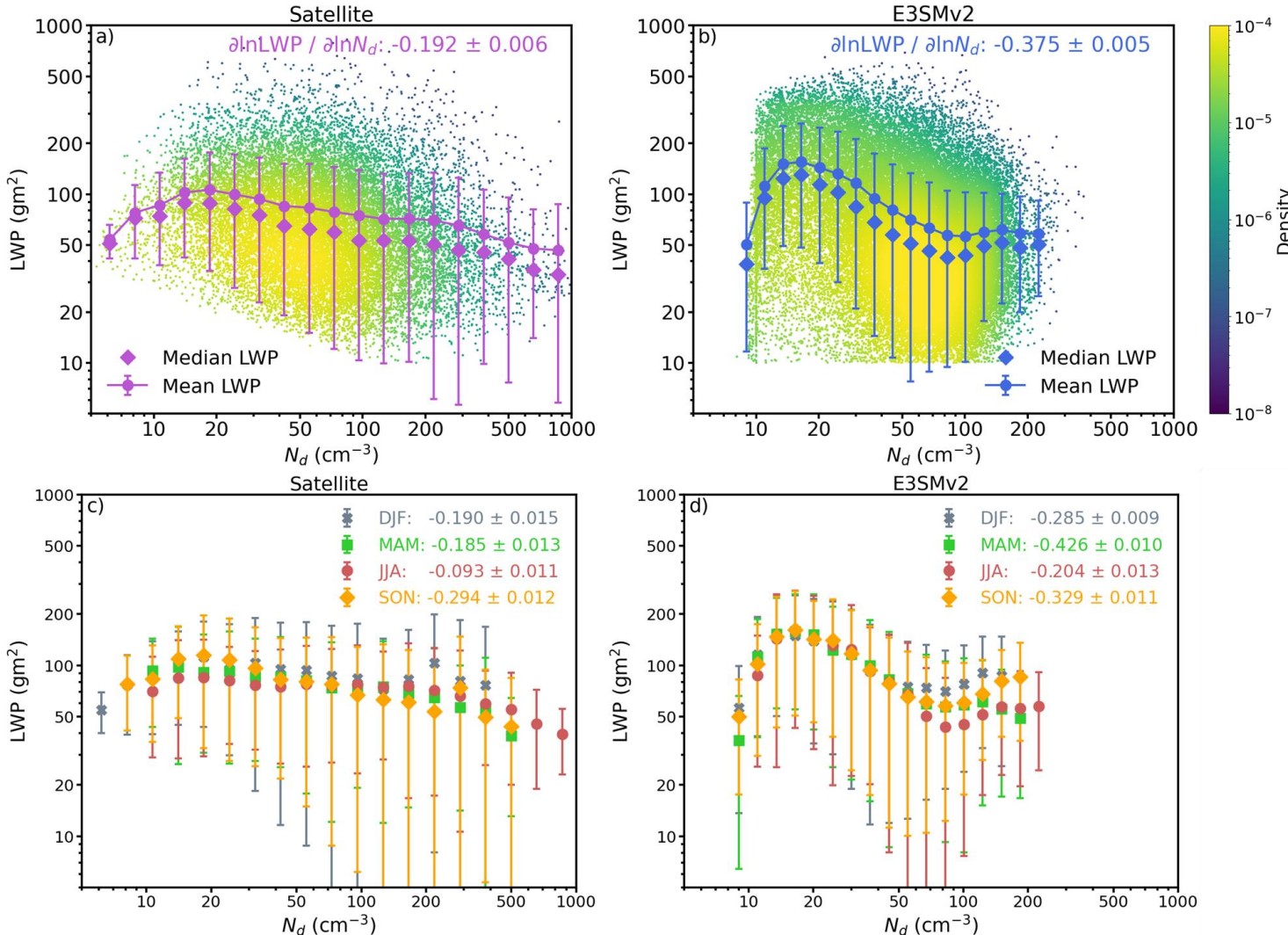

**Figure 2.** Top Panel: Bulk LWP binned as a function of $N_d$ from a) satellite and b) E3SMv2, the Gaussian kernel density estimate (KDE) is shown as color-shaded scatter points area. The mean (median) LWP values in $N_d$ bins are shown as solid-dotted lines (diamond symbols); Bottom panel: seasonality of the LWP dependences from c) satellite and d) E3SMv2, with colored symbols denote mean LWP in $N_d$ bins. Error bars denote standard deviations of LWP in $N_d$ bins. Slopes are ordinary least squares fits of $\partial \ln(LWP)/\partial \ln(N_d)$, and '±' values are standard errors of the fitted slopes.

The dependence of LWP on $N_d$ for both satellite data and E3SMv2 simulations, is presented in
Figure 2. We quantify the response of LWP to changes in $N_d$ using a LWP adjustment index defined as
$\mathcal{L}_0 = \frac{\partial \ln(LWP)}{\partial \ln(N_d)}$ .
We compute $\mathcal{L}_0$ as the slope of an ordinary least squares fit in log-log space between $N_d$ and LWP.
Hence, the $\mathcal{L}_0$ derived from satellite observations and model simulations is -0.192 ± 0.006 and -0.375 ±
0.005, respectively. The '±' values reported are the standard errors (SE) of the slope from that fit
(equivalently, 95% confidence level CI = slope ± 1.96*SE, under standard linear-regression assumptions).
The bulk values are consistent with previous satellite studies over the eastern Atlantic region (Gryspeerdt
et al., 2019; Christensen et al., 2023; Zhang et al., 2025). Both datasets show an inverted-V LWP- $N_d$
relationship. LWP rises with $N_d$ at low $N_d$ then turns negative at higher $N_d$, with a turning point near 20
cm$^{-3}$. In the observations (Fig. 2a), increasing $N_d$ suppresses precipitation and allows LWP to accumulate
at low $N_d$, while at higher Nd the response becomes negative, consistent with enhanced entrainment and
evaporative losses. These results are broadly consistent with prior satellite studies over marine
stratocumulus: an inverted-V LWP-$N_d$ relationship has been reported for $N_d$ ranges of ~10-300 cm$^{-3}$ in
the southeast Pacific (Goren et al., 2025), globally (Gryspeerdt et al., 2019; 2022), and ~7-400 cm$^{-3}$ for
subtropical stratocumulus (Possner et al., 2020). In contrast, the model simulations appear to generate a
similar, yet exaggerated, shape predominantly through parameterized precipitation suppression
(Mülmenstädt and Feingold et al., 2018; Mülmenstädt et al., 2024b).
Global climate models such as E3SMv2 typically lack the resolution to explicitly simulate small-
scale turbulent mixing and entrainment, instead, relying on bulk parameterizations that tend to
overestimate precipitation suppression (Varble et al., 2023; Mülmenstädt et al., 2024a; Y. Zhang et al.,
2024). Noticeably, E3SMv2 (Fig. 2b) captures the relationship between LWP versus $N_d$ qualitatively.
However, the model systematically produces higher LWP at low $N_d$, and lower LWP at high $N_d$ than
observation, suggesting that E3SMv2 overestimate the sensitivity of LWP to $N_d$. This discrepancy may
reflect uncertainties in the parameterization of cloud adjustments, particularly those involving
entrainment and aerosol-cloud microphysical interactions in E3SMv2, as discussed further in Section 4.
The inverted-V shape in the LWP–$N_d$ relationship persists across seasons in both satellite
observations and E3SMv2 simulations (Figs. 2c and 2d), though its intensity and turning point vary.
Satellite observations (Fig. 2c) generally exhibit a weaker negative slope of LWP with $N_d$ during winter
(DJF) and autumn (SON) than in summer (JJA), with transitional behavior in spring (MAM). In contrast,
E3SMv2 (Fig. 2d) show stronger LWP changes with Nd in every season, consistent with the bulk
tendency. Despite the overall similarity in seasonal patterns, the discrepancies between the model and
observations become more apparent when the data are stratified by season.

To gauge how satellite retrieval uncertainty affects the LWP versus $N_d$ relationship, we note that

satellite $N_d$ studies report normalized root-mean-square differences of about 30 to 50% relative to aircraft
data as in Gryspeerdt et al. (2022). In log space a multiplicative $N_d$ error is additive in $\ln N_d$, implying an
expected slope damping of roughly 10 to 30%. Consistent with this expectation, a Monte Carlo test that
multiplied satellite $N_d$ by lognormal noise with coefficient of variation ~30-50% and refit the slope 1000
times reduced the bulk satellite slope from -0.192 to a median of -0.157, an attenuation of about 18%,
with a 95 percent sensitivity band of -0.170 to -0.143. However, this does not alter the sign or the
comparative result reported here, but it does indicate that observed magnitudes should be interpreted with
caution.
**3.3 Vertical distribution of aerosol extinction coefficient**

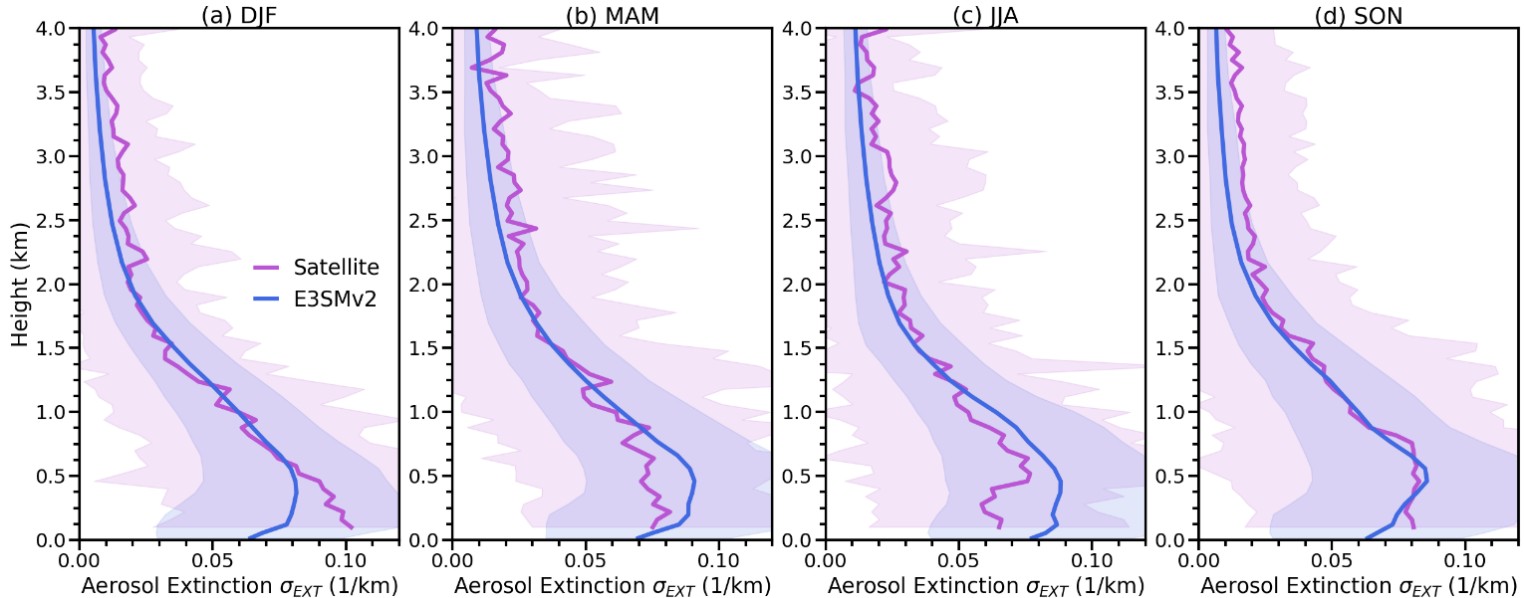

**Figure 3.** Domain averaged vertical distribution of aerosol extinction coefficients ($\sigma_{EXT}$) in the presence of clouds from Satellite (purple) and E3SMv2 (blue) during a) winter, DJF; b) spring, MAM; c) summer, JJA, and d) fall, SON. Shaded area denote the standard deviation of $\sigma_{EXT}$ for each level.

Figure 3 shows seasonal aerosol extinction profiles over the ENA. Both the satellite record and
E3SMv2 feature strongest extinction near the surface below about 1 km with a rapid drop above roughly
2 km. This steep gradient suggests that the aerosols are more concentrated within the MBL, consistent
with surface aerosol sources such as the oxidation of dimethyl sulfide (DMS) and sea spray aerosols
(Zheng et al., 2018; Wang et al., 2021; Ghate et al., 2023). Also, the relatively higher relative humidity
within the MBL might impacts the optical properties of aerosols (Baynard et al., 2006; Feng et al., 2016).
In winter (DJF) and fall (SON), the model underestimates $\sigma_{EXT}$ below 1 km compared to satellite
observations, whereas in spring (MAM) and summer (JJA) it overestimates near-surface extinction.
Recent studies like Logan et al. (2014) and Zheng et al., (2018) demonstrated that the seasonality in the
MBL aerosol properties are highly sensitive to local meteorological conditions and long-range transport
events. During the cold seasons (DJF and SON), the ENA region experiences high wind speeds in the
MBL due to an intensified pressure gradient between the Icelandic Low and the Azores High (Logan et
al., 2014; Ghate et al., 2021). The observed high $\sigma_{EXT}$ in cold seasons reflects coarse-mode contributions
from both enhanced sea-salt emissions under strong MBL winds and episodic Saharan dust intrusions
that reach the ENA via the synoptic northwestward transport (Logan et al., 2014; Gläser et al., 2015;
Rodríguez and López-Darias, 2024). E3SMv2 likely underpredicts this signal due to low sea spray
(Burrows et al., 2020) and an underrepresentation of dust vertical extent and transport (Feng et al., 2022),
a broader model tendency that appears over the North Atlantic as well (H. Wang et al., 2020; Qin et al,

2024).

In warm months (MAM and JJA), ENA is characterized by enhanced formation of secondary
organic aerosols (SOA) and DMS-derived sulfate (Zheng et al., 2018; Sanchez et al., 2018), dominated
by fine-mode aerosols, with weaker sea salt under lower winds and reduced dust transport as the Azores
High strengthens. (Wang et al., 2021). Conversely, the overestimation of $\sigma_{EXT}$ by E3SMv2 may be
partially attributed to the overproduced sulfate and organic matter at the surface (Hassan et al., 2024;
Huang et al., 2024). On the other hand, CALIPSO retrieval limitations under very clean conditions
(Painemal et al., 2019) may also contribute to model-observation differences in the free troposphere.
Overall, E3SMv2 captures the seasonal evolution of the vertical profile and the mean boundary-layer
extinction well enough to support subsequent analysis of simulated cloud responses to aerosol changes.

 **3.3 N$_d$ susceptibility to aerosol extinction coefficient**

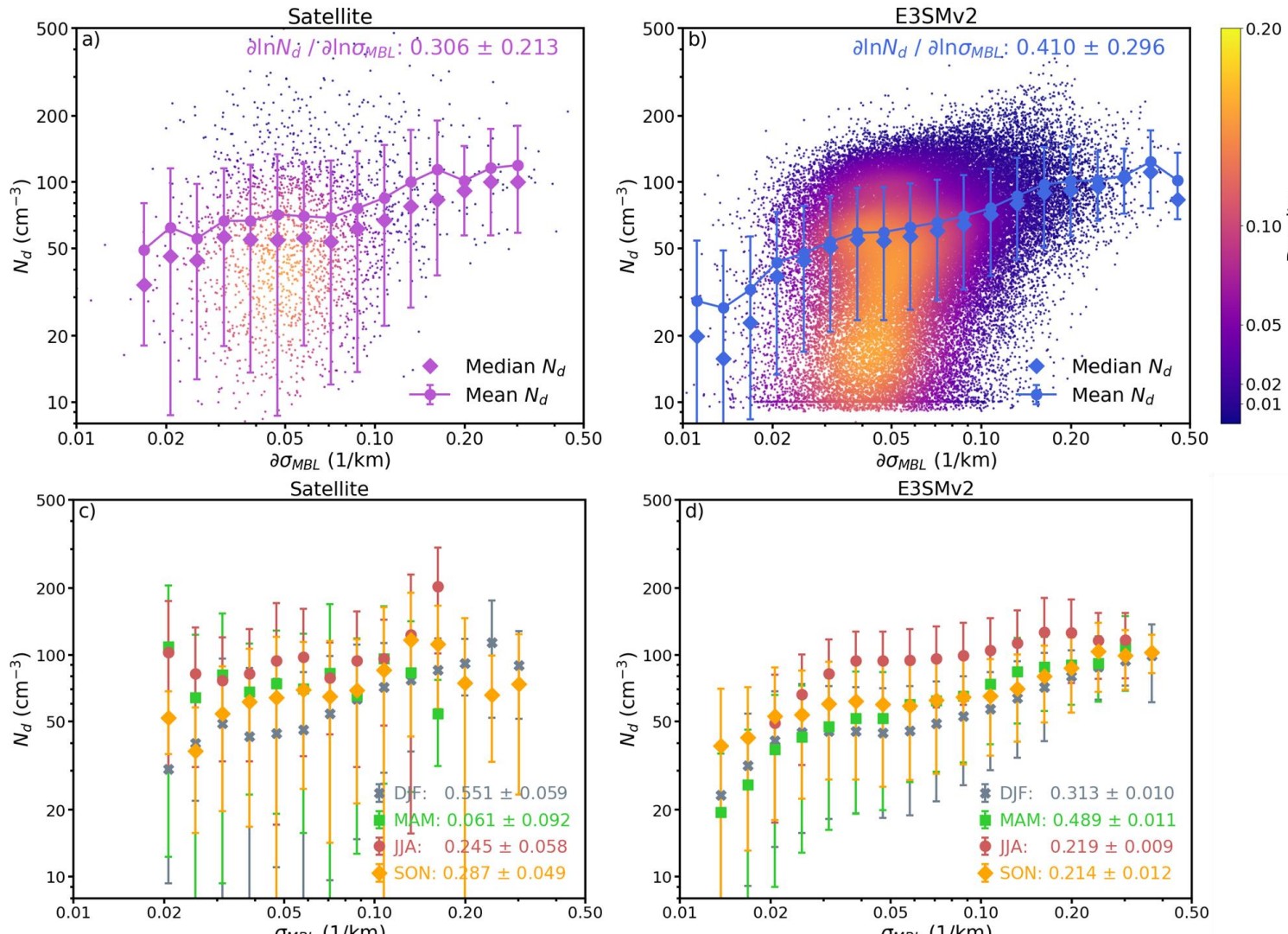

**Figure 4.** Top Panel: $N_d$ binned as a function of $\sigma_{MBL}$ from a) satellites and b) E3SMv2. The Gaussian kernel density estimate (KDE) is shown as color-shaded scatter points area. The mean (median) $N_d$ values in $\sigma_{MBL}$ bins are shown as solid-dotted lines (diamond symbols). Bottom panel: seasonality of the $N_d$ dependences from c) satellite and d) E3SMv2, with colored symbols denote mean $N_d$ in $\sigma_{MBL}$ bins. Error bars denote standard deviations of $N_d$ in $\sigma_{MBL}$ bins. Slopes are ordinary least squares fits of $\partial ln(N_d)/\partial \ln(\sigma_{MBL})$, and '±' values are standard errors of the fitted slopes.

To quantify the aerosol impact on cloud microphysics, we define an aerosol-cloud interaction (ACI) index as:

$$ACI_N = \frac{\partial \ln(N_d)}{\partial \ln(\sigma_{MBL})},$$

where the $\sigma_{MBL}$ denotes the mean value of the below-cloud-top $\sigma_{EXT}$. This parameter represents the sensitivity of $N_d$ to changes in aerosols within the marine boundary layer (MBL). Aircraft in situ measurements near cloud base provide the most physically robust ACI assessment (Gupta et al., 2021; Zheng et al., 2024). However, it is challenging to do that with satellite data and model outputs, because satellite remote sensing like CALIOP cannot reliably determine cloud-base height, and the model's coarse vertical resolution makes it difficult to collocate the model cloud-base with CALIOP layers. Hence, those factors necessitate the use of the mean aerosol properties within the below-cloud-top MBL in the present study, facilitating a consistent comparative assessment of aerosols between satellite observations and model simulations.

The top row of Figure 4 compares the satellite-derived and E3SMv2-simulated relationships between $N_d$ and $\sigma_{MBL}$. Satellites (Fig. 4a) show a moderate $N_d$ sensitivity with considerable scatter ($ACI_N = 0.306 \pm 0.213$). The positive $ACI_N$ reflects the Twomey effect and lies within reported satellite ranges over marine stratocumulus and the eastern Atlantic. For example, McCoy et al. (2017) found a log-log slope of 0.31 between $N_d$ and sulfate mass, Jia et al. 2021 reported 0.14-0.51 for $N_d$ versus AOD over oceans, and recent reviews summarize satellite-based susceptibilities of about 0.1–0.7 depending on sampling and aerosol proxies (Gryspeerdt et al., 2023). In comparison, E3SMv2 (Fig. 4b) reproduces the qualitative increase of $N_d$ with $\sigma_{MBL}$, but with a noticeably steeper slope, consistent with an overly sensitive microphysical or activation response noted in previous studies (Christensen et al., 2023; Varble et al., 2023). In other words, for the same fractional change in $\sigma_{MBL}$, E3SMv2 predicts a larger fractional change in $N_d$ compared to satellite data. This model–observation discrepancy may reflect uncertainties in how E3SMv2 parameterizes aerosol activation, updraft velocities at the cloud base, or boundary-layer processes such as entrainment and mixing (Tang et al., 2024; Wan et al., 2025), as discussed further in Section 4.

In terms of seasonal variations in $N_d$ susceptibility, Satellite derived $ACI_N$ (Fig. 4c) is largest in DJF near 0.55 and smallest in MAM near 0.06, with JJA and SON intermediate. E3SMv2 (Fig. 4d) remains positive in all seasons but spans a narrower range, including a much stronger MAM value near 0.49 than observed and a weaker DJF response than satellites, while JJA and SON are closer to observations, yet the model maintains a higher and more uniform sensitivity overall.

Taken together, E3SMv2 tends to produce both stronger LWP responses to $N_d$ and enhanced $N_d$
responses to aerosol than satellite derived relationships. A notable limitation of this seasonal grouping is
that it fails to disentangle the complexities of aerosol-cloud interactions from characteristic
meteorological variations. Aggregating data into seasons makes it challenging to unambiguously
attribute changes in cloud properties to aerosol variations rather than to shifts in large-scale dynamics or
thermodynamic conditions (Zheng et al., 2022b; Zhang et al., 2023). This limitation necessitates the
regime-based analysis in Section 4, in which samples are clustered in dominant meteorological regimes
and can more effectively isolate microphysical processes from the confounding effects of synoptic-scale
and seasonal variability (Mülmenstädt et al., 2012; Mechem et al., 2018; Zheng et al., 2025).

## 393 4 Regime-based Analysis of Aerosol, cloud properties, and their interactions

## 394 4.1 Distinctive Meteorological Regimes over ENA

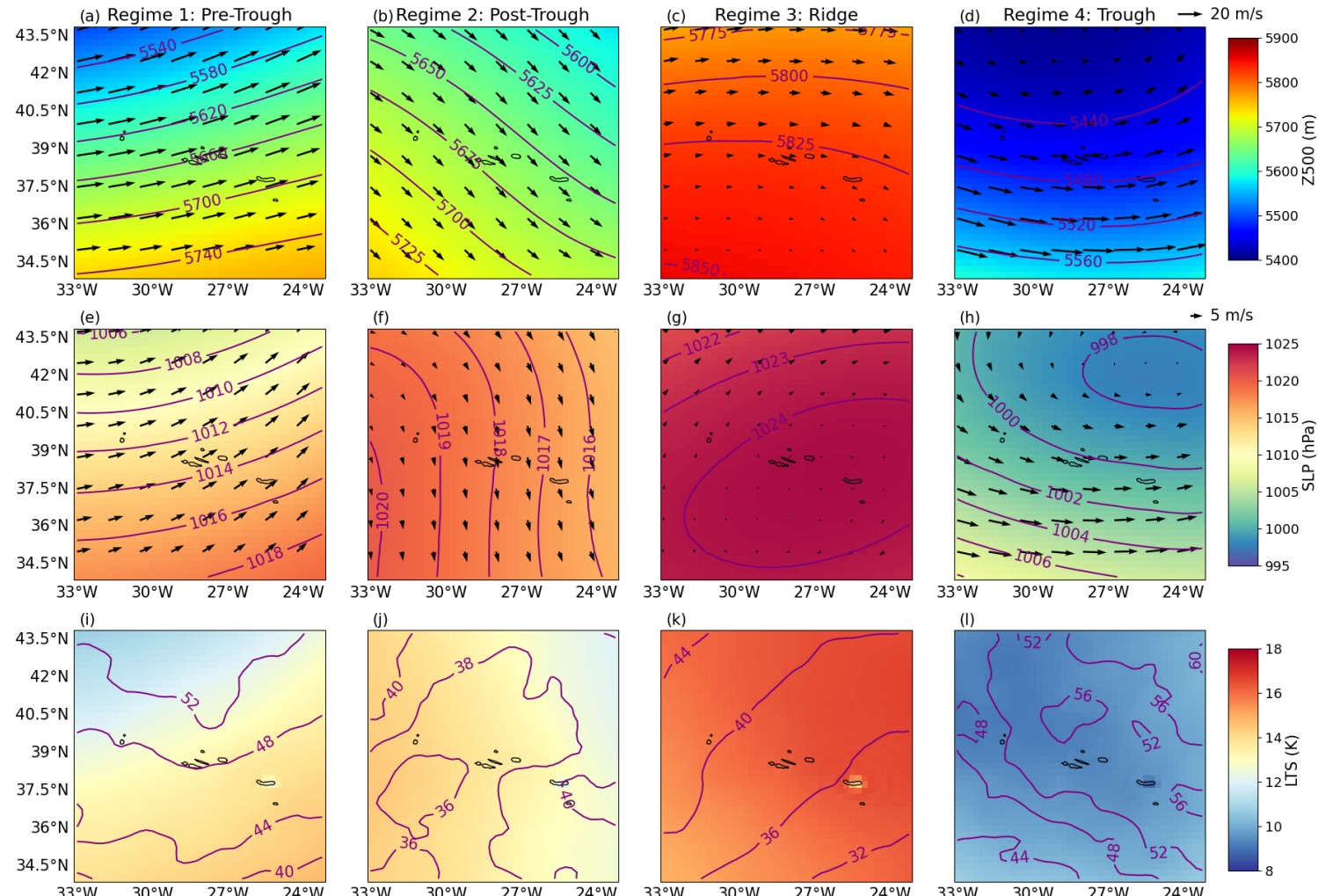

**Figure 5.** Meteorological composites for each synoptic regime classified by the clustering model: Regime 1 (a, e, i), Regime 2 (b, f, j), Regime 3 (c, g, k), and Regime 4 (d, h, l). The first row shows 500 hPa geopotential height (Z500) shaded and contoured, with 500 hPa wind vectors overlaid. The second row presents sea level pressure (SLP) in both shaded and contoured formats, together with 10 m surface wind vectors. The third row displays lower tropospheric stability (LTS) at 700 hPa as the shaded field and 700 hPa relative humidity (RH700) in contours.

As detailed earlier, the CNN–LSTM–DEC clustering of 3,286 daily ERA5 states results in the
identification of four distinct synoptic-scale regimes (Figure 5). Namely, Pre-Trough (regime 1), Post-
Trough (regime 2), Ridge (regime 3) and Trough (regime 4). For each regime, composites were computed
as the arithmetic mean of the corresponding ERA5 fields across all time steps assigned to that regime.
Regime 1 (Figs. 5a, 5e, and 5i) represents the Pre-Trough phase, characterized by a developing
trough approaching the Azores. This regime features strong southwesterly winds at both 500 hPa and
near the surface, moderate mid-level moisture, and low lower tropospheric stability (LTS). Such
conditions typically precede frontal development and passage and are associated with the early stages of
midlatitude cyclone progression, therefore resulting in a wet free troposphere and low LTS (Mechem et
al., 2018; Zheng et al., 2025).
Regime 2 denotes the typical Post-Trough condition that follows the passage of a trough,
characterized by prevailing northwesterly winds (Fig. 5b) and transitional stability. The SLP field reveals
a relatively weak pressure gradient, corresponding to a post-frontal environment in which drier and colder
air is advected into the region (Figs. 5f and 5j). Although LTS remains moderate, it is slightly higher
than in Regime 1, reflecting the gradual stabilization behind the frontal system. Taken together, Regimes
1 and 2 depict the typical meteorological evolution associated with mid-latitude cyclones traversing the
ENA, which occur regularly throughout the year (Table 2), particularly during colder seasons when mid-
latitude cyclone activity is more frequent (Wood et al., 2015; Mechem et al., 2018).
Regime 3 corresponds to the Ridge phase, in which a pronounced ridge dominates the region.
Both the SLP and 500 hPa geopotential height (Z500) fields constitute a broad anticyclonic pattern, with
relatively weak and variable winds from the surface up to 500 hPa. This pattern coincides with the driest
free troposphere and the most stable lower troposphere observed among all regimes, which generally
favor a more coupled and shallower (MBL) (Carrillo et al., 2015; Zheng et al., 2025). As indicated in
Table 2, Regime 3 is the most frequently occurring regime (63.8%) in the region, peaking during the
summer months when the center of the Azores High predominantly lies to the southwest of the Azores
Islands. This finding is consistent with previous studies (Mechem et al., 2018; Wang et al., 2022), and
the synoptic categorization is similar to that in Painemal et al. (2023) for the Western North Atlantic.

Regime 4 represents a typical Trough phase, characterized by a canonical 500 hPa trough with

stronger cyclonic flow at the surface. The lower troposphere exhibits reduced static stability, while the
contours of relative humidity at 700 hPa (RH700, Fig. 4l) indicate a more humid troposphere. These
conditions imply enhanced ascent and moist processes. Note that among the four regimes, Regime 4 is
the least frequent (3.4%) and is largely confined to the colder seasons (winter and spring), confirming
the findings from previous studies (Wood et al., 2015; Mechem et al., 2018; Wang et al., 2022).
**4.2 Cloud properties under different regimes**

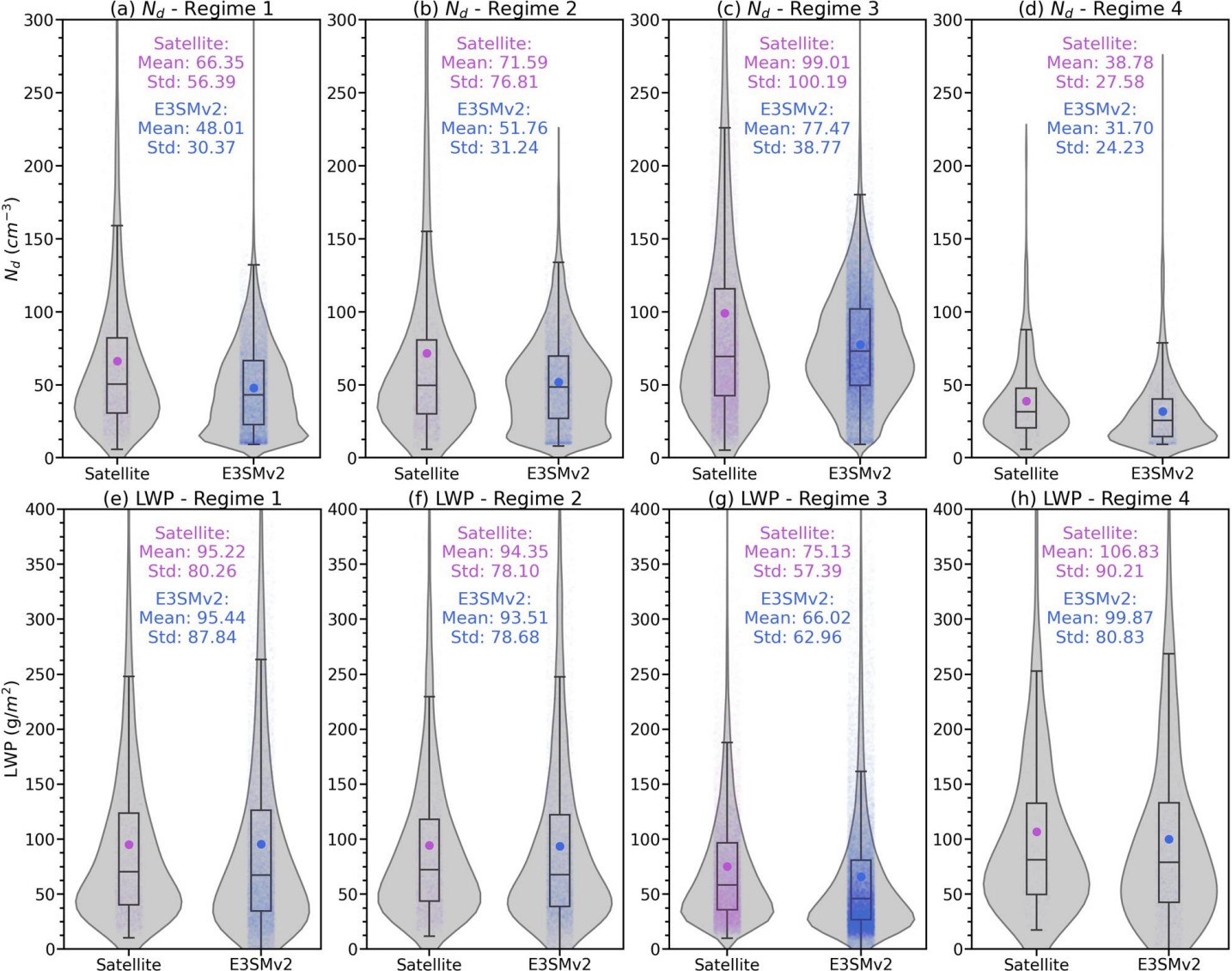

**Figure 6.** Violin plots of cloud droplet number concentration ($N_d$, top panels) and cloud liquid water path (LWP, bottom panels) from satellite retrievals (purple) and E3SMv2 simulations (blue), grouped by Regime 1 (a, e), Regime 2 (b, f), Regime 3 (c, g), and Regime 4 (d, h). The mean value is indicated by the color-coded dot. The smoothed shape of each violin shows the Gaussian kernel density estimate (KDE). From top to bottom within each violin, the box plot lines represent the third quartile (Q3, 75th percentile), median (Q2, 50th percentile), and first quartile (Q1, 25th percentile), respectively. The upper whisker extends to Q3 + 1.5 × IQR (IQR: interquartile range), and the lower whisker extends to Q1 − 1.5 × IQR.

In order to provide synergy on the meteorology and cloud for reference in this study, a brief and
qualitative summary of meteorology patterns and cloud and precipitation status are listed in Table 1.
Furthermore, based on the four distinct meteorological regimes identified through clustering, we stratify
cloud properties from both satellite retrievals and E3SMv2 simulations. Both $N_d$ and LWP exhibit
systematic regime-dependent behavior and model biases (Fig. 6). And the detailed quantities are listed
in Table 3.
Across all regimes, E3SMv2 tends to underestimate $N_d$ while representing LWP more accurately.
A rigorous comparison of surface precipitation rates between satellite data and E3SMv2 is limited in this
study due to lack of collocated precipitation rate measurements from the satellite. Therefore, we choose
to compare the in-cloud fractional occurrences of rain and rain LWP from E3SMv2 with the fractional
occurrences of drizzle, light rain, and rain from CloudSat (Table S1). In addition, CTH statistics,
stratified by regime from both satellite and E3SMv2 datasets, are presented in Table 2, with composite
maps shown in Figures S1 and S2. To quantify the variability of CTH under different regimes, Moran's
I indices, a measure of spatial autocorrelation, were computed and then normalized for each regime to
account for the differences in spatial resolution between the satellite and E3SMv2 datasets. A normalized
Moran's I index of 0 means the least autocorrelated, and 1 means the most autocorrelated. The normalized
Moran's I indices for the satellite (E3SMv2) datasets are 0.61 (0.0), 0.64 (0.89), 1.0 (1.0), and 0.0 (0.13)
for Regimes 1, 2, 3, and 4, respectively.
Regime 1, characterized by an approaching trough with southwesterly flow, moderate moisture,
and relatively weak subsidence, exhibits a mean $N_d$ of 66.35 cm⁻³ in satellite observations, whereas
E3SMv2 simulates a lower mean $N_d$ of 48.01 cm⁻³ with reduced variability. The mean LWP in E3SMv2
(95.44 g m⁻²) is similar to that of the satellite retrievals (95.22 g m⁻²), with a slightly broader distribution.
These findings suggest that while E3SMv2 captures the overall liquid water content in cloud layers under
pre-frontal conditions, it systematically underestimates $N_d$. Moreover, the CTH in Regime 1 remains low
(Table 3), reflecting a shallower MBL in the transitional environment of the approaching front (Jeong et
al., 2022). Precipitation statistics indicate moderate fractions of drizzle and light rain (Table S1),
consistent with the notion that although pre-trough instability favors drizzle production, the boundary
layer does not fully deepen to support frequent or intense rainfall (Wood, 2005; Wu et al., 2020; Zheng
et al., 2022b). The partial uplift and moderate moisture convergence can occasionally enhance cloud
thickness, as indicated by a moderately high normalized Moran's I index (0.61), yet the overall shallower
structure typically limits heavier precipitation.

In the post-trough environment of Regime 2, the northwesterly flow advects drier and cooler air

behind the frontal system. Such conditions are typically associated with increased subsidence and a
deeper MBL conducive to the development of stratocumulus clouds (Wu et al., 2020; Jensen et al., 2021;
Jeong et al., 2022). Consequently, satellite-derived CTH values (Table 3 and Fig. S1b) are higher than
those in Regime 1, indicating a deeper MBL. Furthermore, clouds in Regime 2 are also associated with
the highest fractional occurrences of drizzle among the four regimes (Table S1). Drizzle formation and
turbulent mixing can lead to heterogeneous cloud structures with less cloud adiabaticity (Wu et al., 2020),
as reflected by a normalized Moran's I index of 0.64 and in Figure S1b, in contrast to the more uniform
CTH variation in Regime 3. E3SMv2 captures the relative CTH variation and precipitation frequency
reasonably well under this regime. As for Regime 1, E3SMv2 still systematically underestimates $N_d$
(51.76 cm⁻³ vs. 71.59 cm⁻³) while simulating the observed mean LWP well (93.51 vs. 94.35 g m⁻²).

Regime 3 features with a pronounced ridge, reduced convective activity, and generally lower

moisture in the boundary layer. Satellite retrievals show the highest mean $N_d$ (99.01 cm⁻³, Fig. 6c), which
could be attributed to the less active drizzle processes and droplet evaporation in shallower cloud deck
within a more stable MBL (Wood et al., 2012; Zheng et al., 2024). In contrast, E3SMv2 simulates a
lower mean $N_d$ (77.47 cm⁻³) and fails to capture the broad observed distribution. Moreover, LWP in this
ridge regime is the lowest among all regimes, with E3SMv2 underestimating satellite observations (66.02
vs. 75.13 g m⁻²; Fig. 6g). From both satellite and model perspectives, the cloud field is characterized by
shallower, more homogeneous decks that span large horizontal areas yet produce only light or sporadic
drizzle, as reflected by lower CTH (Figs. S1c and S2c) and the lowest precipitation fractions among the
four regimes (Table S1). Overall, the results in Regime 3 align with the signature of shallow stratus and
stratocumulus clouds (Rémillard and Tselioudis, 2015; Mechem et al., 2018; Wu et al., 2020; Jensen et
al., 2021).

Under a well-developed trough in Regime 4, satellite observations record the lowest $N_d$

(38.78 cm⁻³) but the highest LWP (106.83 g m⁻²) among four regimes, reflecting the prevalence of deep

and warm-rain-active cloud systems formed by strong uplift and abundant moisture. These conditions yield the highest and most variable CTH among the four regimes (Table 3), along with frequent precipitation in the form of drizzle or rain. Stronger vertical motion promotes the development of deeper clouds with higher rainfall efficiency, contributing to spatially heterogeneous precipitating cloud fields observed in both the satellite data and model simulations (Figs. S1d and S2d; Table S1). Notably, E3SMv2 simulates the high LWP (99.87 g m$^{-2}$) and elevated liquid water content as in the observations, but underestimates variability. Similarly, the model underestimates $N_d$ (31.70 cm$^{-3}$) relative to satellite observations as in the other regimes.

Overall, these results demonstrate the clear meteorological impact on both cloud microphysical ($N_d$) and macrophysical (LWP) properties. Pre- and post-trough conditions (Regimes 1 and 2) favor moderate $N_d$ and LWP, while ridge-dominated conditions (Regime 3) promote stable, stratiform-dominated environments with high mean $N_d$ but relatively low LWP. In contrast, developed troughs (Regime 4) yield lower $N_d$ yet substantially higher LWP in more vertically developed cloud systems. Although E3SMv2 captures the LWP mean and distributions across these synoptic regimes, it systematically underestimates $N_d$. This discrepancy suggests that the challenges in simulating cloud droplets are irrespective of the meteorological influences.

**Table 1. Summary of meteorological and cloud categories in different regimes.**

| Regimes | Meteorological Patterns | Cloud Status |
|---|---|---|
| **R1 Pre-Trough** | Approaching trough; strong SW winds at 500 hPa & surface; moderate mid-level moisture; low LTS | Lower CTH; moderate drizzle / light rain |
| **R2 Post-Trough** | Post-frontal NW winds; moderate LTS; weak pressure gradient; drier, free troposphere | Higher CTH; high drizzle fraction but low overall precipitation |
| **R3 Ridge** | Broad anticyclonic ridge; high-pressure-dominated surface; driest free troposphere; highest LTS | Shallow MBL; Lowest CTH; minimal precipitation |
| **R4 Trough** | Canonical trough; strong cyclonic surface flow; lowest LTS; moist free troposphere | Highest & most variable CTH; most frequent drizzle / rain |

 **Table 2. Regime occurrences over ENA at 13:00 LT (1 p.m. local time) from ERA5 inputs**

| Count (Fraction %)* | Regime 1 Pre-Trough | Regime 2 Post-Trough | Regime 3 Ridge | Regime 4 Trough |
|---|---|---|---|---|
| **Winter (DJF)** | 269 (33.2%) | 166 (20.4%) | 313 (38.6%) | 63 (7.8%) |
| **Spring (MAM)** | 193 (23.3%) | 169 (20.4%) | 424 (51.2%) | 42 (5.1%) |
| **Summer (JJA)** | 26 (3.1%) | 22 (2.7%) | 780 (94.2%) | 0 (0%) |
| **Fall (SON)** | 111 (13.6%) | 122 (14.9%) | 581 (70.9%) | 5 (0.6%) |
| **Total** | 599 (18.2%) | 479 (14.6%) | 2098 (63.8%) | 110 (3.4%) |

*Percentages in bracket denote the fractional occurrence of specific regime among total sample in each season category (and total sample).

**Table 3. Regime-based aerosol and cloud variables from Satellite and E3SMv2**

| | Regime 1 | Regime 2 | Regime 3 | Regime 4 |
|---|---|---|---|---|
| **Satellite** | | | | |
| $N_d$ $(cm^{-3})$ | $66.35 \pm 56.39$ | $71.59 \pm 76.81$ | $99.01 \pm 100.19$ | $38.78 \pm 27.58$ |
| $LWP$ $(gm^2)$ | $95.22 \pm 80.26$ | $94.35 \pm 78.10$ | $75.13 \pm 57.39$ | $106.83 \pm 90.21$ |
| $CTH$ $(km)$ | $1.59 \pm 0.46$ | $1.76 \pm 0.42$ | $1.46 \pm 0.50$ | $1.70 \pm 0.45$ |
| $\sigma_{MBL}$ $(1/km)$ | $0.075 \pm 0.047$ | $0.070 \pm 0.046$ | $0.063 \pm 0.039$ | $0.067 \pm 0.048$ |
| **E3SMv2** | | | | |
| $N_d$ $(cm^{-3})$ | $48.01 \pm 30.37$ | $51.76 \pm 31.24$ | $77.47 \pm 38.77$ | $31.70 \pm 24.23$ |
| $LWP$ $(gm^2)$ | $95.44 \pm 87.84$ | $93.51 \pm 78.68$ | $66.02 \pm 62.96$ | $99.87 \pm 80.83$ |
| $CTH$ $(km)$ | $1.62 \pm 0.84$ | $1.76 \pm 0.68$ | $1.41 \pm 0.59$ | $1.81 \pm 0.88$ |
| $\sigma_{MBL}$ $(1/km)$ | $0.070 \pm 0.038$ | $0.061 \pm 0.033$ | $0.071 \pm 0.041$ | $0.063 \pm 0.032$ |

 **4.3 LWP-N$_d$ relationships under different regimes**

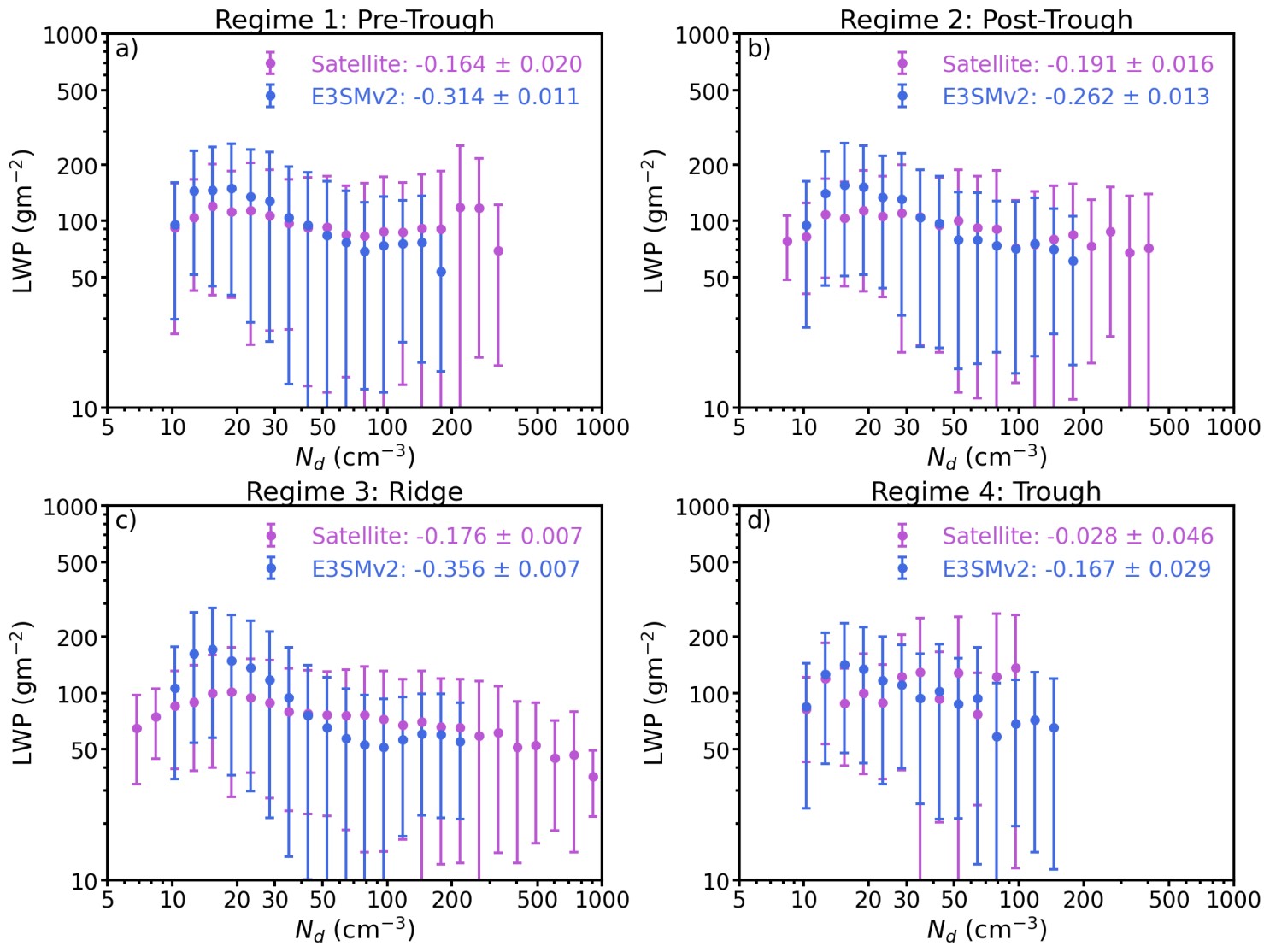

**Figure 7.** LWP responses on $N_d$ from satellite (purple) and E3SMv2 (blue) for a) Regime 1, b) Regime 2, c) Regime 3, and d) Regime 4. Colored dots denote mean LWP in $N_d$ bins, and whiskers denote standard deviations. The quantitative LWP adjustment index $\mathcal{L}_0 = \partial\ln(LWP)/\partial\ln(N_d)$ is denoted in the legend, and '$\pm$' values are standard errors of the fitted slopes.

The relationship between LWP and $N_d$ across the four synoptic regimes in both satellite retrievals

and the E3SMv2 model exhibits the characteristic inverted-V shape (Fig. 7), as shown in the seasonal
assessment in Section 3. Furthermore, regime-specific meteorological differences, particularly variations
in stability and moisture transport, strongly influence the shape and peak of the LWP–$N_d$ relationship.
Across regimes, the peak in the satellite data of the inverted LWP-$N_d$ curve occurs at $N_d \approx$ 15-96 cm⁻³
(particularly, 96.5 cm⁻³ at Regime 4), and at LWP $\approx$ 101-136 g m⁻²; whereas the E3SMv2 peaks at a
much narrower $N_d \approx$ 15-19 cm⁻³ with higher peak LWP $\approx$ 142-171 g m⁻². The quantitative LWP
adjustment ($\mathcal{L}_0$) values are listed in Table 4.

Noticeably, satellite retrievals in the Ridge regime (R3) display a more pronounced inverted-V

shape compared to themselves in other regimes, with LWP consistently declining as $N_d$ increases at high
values (Fig. 7c). Under Ridge conditions, strong subsidence limits vertical growth, reduces moisture
convergence, and produces a shallower, more stable cloud layer. Although the mixing rate from
entraining drier free-tropospheric air might not be as intense as in thicker and more turbulent clouds, the
drying effect propagates more efficiently through the thinner cloud layer, exerting a significant impact
(Sanchez et al., 2020; Chun et al., 2023). Furthermore, since the Ridge regime features more high-$N_d$
conditions, the resulting smaller cloud droplets are more susceptible to entrainment-evaporation, leading
to a more efficient removal of cloud water (Possner et al., 2020). In contrast, the E3SMv2 model
significantly overestimates the maximum LWP, and yields a significantly greater LWP decline with $N_d$
($\mathcal{L}_0$ values of -0.356 for model vs. -0.176 for satellite).

Under Regime 1 (Pre-Trough, Fig. 7a), the satellite and model data show a rise in LWP as $N_d$

increases from very low values, while the characteristic inverted-V shape is partially masked by increased
scatter in the satellite-retrieved LWP at high $N_d$, likely due to the complexity of drizzle, entrainment, and
mixing processes in the pre-frontal environment. The enhanced drizzle formation, indicated by the
relatively higher drizzle and rain fractions in clouds, can reduce the temperature gradient near cloud-top
via condensation warming, stabilizing the cloud layer and partially offsetting the entrainment cooling
and retaining the LWP. Similar to those in Regime 3, the model also exhibits a nearly double $\mathcal{L}_0$ (-0.314)
compared to satellite (-0.164). One possible explanation is that the model microphysics suppresses
drizzle too aggressively when $N_d$ increases occur at lower values (Varble et al., 2023; Mülmenstädt et al.,
2024b), thereby promoting the accumulation of cloud liquid water prior to the onset of entrainment-
drying or precipitation. Furthermore, the MG2 microphysics scheme may trigger the entrainment–
evaporation feedback too rapidly in the model, resulting in an LWP adjustment timescale much shorter
than observed (Zhou et al., 2025). As a result, the model depletes cloud liquid water too quickly (Xie et
al., 2018; Rasch et al., 2019). These combined model uncertainties lead to an excessively steeper $\mathcal{L}_0$
compared to satellite observations.

In the Post-Trough regime (Regime 2; Fig. 7b), the residual dynamic forcing from the trough coupled with the onset of entrainment-induced evaporation, produces an intermediate LWP–$N_d$ sensitivity, particularly under the high $N_d$ condition, between the Pre-Trough and Ridge regimes. On the one hand, in the Post-Trough environment, the cloud field is more variable and relatively thicker, which is found to be favorable for stronger entrainment rates (Wood, 2007; Wu et al., 2020; Chun et al., 2023), hence depleting LWP. On the other hand, this effect might be closely intermingled with patches of high LWP sustained by the precipitation-stabilization effect (Possner et al., 2020; Wu et al., 2020). These competing effects result in an averaged LWP decline with increasing $N_d$ that is stronger than in the Pre-Trough state but remains less pronounced than the sharp decrease observed in the Ridge regime. Compared to the satellite observations, E3SMv2 underestimates LWP at high $N_d$, coupled with the slightly steeper LWP–$N_d$ slope. The model bias likely results from parameterized entrainment and in-cloud mixing (Y. Zhang et al., 2023), which may intensify cloud-top evaporation cooling and promote LWP loss; in contrast, real clouds exhibit more heterogeneous mixing that better preserves liquid water at high $N_d$.

In Regime 4 (Fig. 7d), characterized by a well-developed trough with enhanced ascent and high moisture availability, clouds deepen, and LWP peaks at relatively high $N_d$, as observed by satellites. The muted LWP decrease seen in satellite data may be largely a consequence of combined factors. Since the satellite $N_d$ is derived assuming an adiabatic vertical profile and a constant $N_d$ throughout the cloud, the expected increase in subadiabaticity for Regime 4, as suggested by the enhanced precipitation, might induce retrieval bias that dampens the apparent sensitivity of LWP to $N_d$ (Grosvenor et al., 2018). At the same time, retrievals tend to average over heterogeneous cloud fields in the Trough regime, where cloud properties vary on small scales, further smoothing out the true microphysical sensitivity (Gryspeerdt et al., 2022). Excluding the retrieval-induced bias, strong updrafts in this regime can promote moisture convergence, which in turn maintain or even enhance LWP despite higher $N_d$ (Goren et al., 2018; Zhang et al., 2022; Painemal et al., 2023). In some cases, heavy precipitation may also drive locally precipitation-generated cold pooling that enhances the cloud base updraft that helps maintain LWP following precipitation. Such muted LWP response to $N_d$, has been reported for the stratocumulus clouds over the southeast Pacific, particularly in thickening stratocumulus clouds with a higher likelihood of producing more intense precipitation (Smalley et al., 2024), suggesting the precipitation-suppressing overwhelms the entrainment-drying effect on LWP. In contrast, E3SMv2 maintains an evident inverted-V shape of LWP–$N_d$ curve as in other regimes, albeit with a smaller $\mathcal{L}_0$ (-0.167), resulting from the explicitly parameterized microphysical processes.

Overall, the model simulations exhibit an excessively sharp rise-and-fall pattern in LWP, producing an
exaggerated inverted-V in the LWP–$N_d$ relationship, compared to the more subtle shapes in satellite
results. This suggests that microphysical feedback to cloud water may be triggered too early and too
rapidly in the model. Such behavior may originate from the MG2 microphysics scheme's nonlinear
autoconversion rate (Gettelman and Morrison, 2015), which acts to suppress drizzle too aggressively at
low $N_d$, thereby retaining excess LWP, but truncates liquid water accumulation as $N_d$ increases (Wang et
al., 2023; Ovchinnikov et al., 2024). These limitations hinder E3SMv2's ability to capture observed cloud
feedback, particularly in regimes with shallower clouds (e.g., Pre-Trough and Ridge). Addressing these
issues may require recalibrating autoconversion rates using observational constraints (e.g., ARM data)
and improving scale-aware entrainment schemes that better differentiate mixing regimes.
**4.4 $N_d$ susceptibility to aerosols under different regimes**

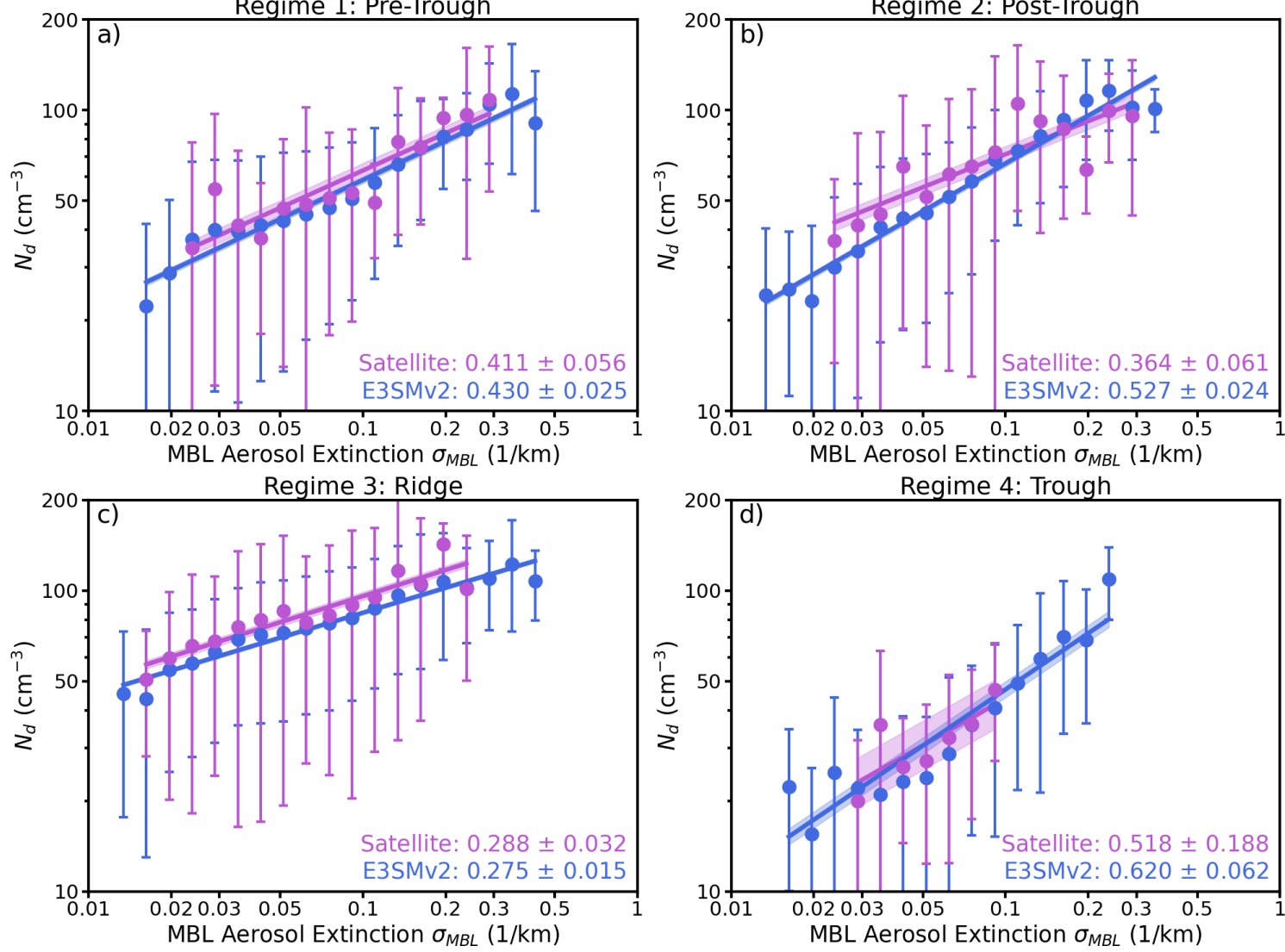

**Figure 8.** Cloud droplet number concentrations ($N_d$) dependence on the mean MBL aerosol extinction coefficient ($\sigma_{MBL}$) from satellite retrievals (purple) and E3SMv2 simulation (blue), under a) Regime 1; b) Regime 2; c) Regime 3 and d) Regime 4. The solid line indicates the regression line, and the $N_d$ susceptibility $ACI_N = \partial \ln (N_d)/\ln (\partial \sigma_{MBL})$ is denoted in the legend, and '±' values are standard errors of the fitted slopes.

Figure 8 illustrates how $N_d$ responds to changes in MBL aerosol extinction ($\sigma_{MBL}$) across each
meteorological regime, as quantified by the aerosol–cloud interaction index ($ACI_N$). Under all regimes,
both observations and E3SMv2 simulations show that $N_d$ generally increases with increasing $\sigma_{MBL}$,
reflecting the typical ACI in which cloud droplet concentration rises with increased aerosol loading. And
it is worth noting that the model is able to simulate the quantitative relationship between $N_d$ and $\sigma_{MBL}$,
with exception of Regime 2. The $ACI_N$ values are listed in Table 4. Since each regime is characterized by
distinct large-scale meteorological conditions, these environmental factors influence the ACI remarkably,
as the aerosol activation is highly influenced by updraft strength, moisture availability, and in-cloud
supersaturation (Chen et al., 2011; Kirschler et al., 2022; Zheng et al., 2024).
Both satellite retrievals and E3SMv2 yield the lowest $N_d$ susceptibility under the Ridge scenario
(Regime 3, Fig. 8c), consistent with the stabilizing effect of subsidence that suppresses updraft variability
and limits cloud depth. Furthermore, solar heating on cloud top, especially prominent in the local
afternoon, can offset the longwave radiative cooling, thereby stabilize the cloud layer and reduce in-
cloud supersaturation (Wood, 2012; Zheng et al., 2018), hence dampening the $N_d$ susceptibility to
aerosols. Such mechanism can exert a larger influence on Regime 3, as it is dominated by the warm
season (summer and fall) occurrences. The good agreement between the model and satellite observations
under these steady conditions suggests that, when cloud-top mixing and convective vigor are limited,
E3SMv2 aerosol activation and microphysics perform reasonably well.
Conversely, under trough conditions (Regime 4), retrievals and E3SMv2 produce the highest $N_d$
susceptibility among all regimes (Fig. 8d). In these conditions, stronger updraft and abundant moisture
produce deeper MBL clouds, and effectively increase the in-cloud supersaturation (Gong et al., 2023).
Therefore, the environmental conditions and the relatively less $N_d$ provide aerosols a greater opportunity
to modulate droplet numbers (Hudson and Noble, 2014; Zheng et al., 2024). The E3SMv2 exhibits a
steeper slope than that observed, suggesting that the model may overestimate the efficiency with which
additional aerosol activates into new droplets, though the discrepancy is less pronounced than those in
Regime 2.

Interestingly, the largest overestimation of $ACI_N$ by E3SMv2 occurs under Post-Trough
conditions (Regime 2, Fig. 8b). In this regime, the model exhibits a much higher $ACI_N$ than observations,
suggesting that the model either overestimates aerosol activation or underrepresents processes that limit
droplet concentration, such as drizzle formation and entrainment mixing. In thicker MBL clouds, the
greater vertical extent may allow for enhanced droplet recirculation and collision–coalescence, resulting
in a reduction of $N_d$ that dampens aerosol effects (O et al., 2018; Zheng et al., 2024). Under Pre-Trough
conditions (Regime 1, Fig. 8a), the model slightly overestimates $N_d$ susceptibility yet agrees better with
observations compared to Post-Trough. Previous studies have shown that E3SMv1 exhibits greater
sensitivity of $N_d$ to aerosols than do ground-based and satellite observations (Christensen et al., 2023;
Varble et al., 2023), and this issue appears to persist in E3SMv2 over the ENA, consistent with Huang et
al. (2024). It is possible that the model does not accurately represent the postfrontal boundary layer,
possibly due to the unresolved subgrid turbulence (Ma et al., 2022), which may lead to an exaggerated
$N_d$ response to aerosol extinction.

Note that under low MBL aerosol ($\sigma_{MBL}$) conditions, the model yields more occasions of low $N_d$
compared to the satellite, though it can be due to the limited sample sizes in the satellite, yet the $N_d$
increases with $\sigma_{MBL}$ are also more subtle, especially in the Post-Trough and Ridge regimes. Previous
studies suggest that when environmental conditions limit activation and updraft, the resulting droplet
numbers are systematically low (Tang et al., 2023; Varble et al., 2023). In other words, the model
parameterizations lead to under-activation of aerosols in low-aerosol or weak-turbulence regimes. In
contrast, sensitivity experiments by Wan et al. (2025) show that while stronger updrafts can boost $N_d$,
doing so would undesirably increase the effective radiative forcing. Therefore, the model aerosol
activation scheme might be oversensitive to the environmental factors, as shown in the present study. It
is also noteworthy that E3SMv2 simulates a greater extent of $\sigma_{MBL}$ than is observed by satellites under
all scenarios, particularly in Regime 4, where precipitation is more prevalent. This discrepancy may arise
from insufficient drizzle scavenging at moderate to high aerosol loads (Shan et al., 2024), whereby the
model fails to effectively remove aerosols, rendering it overly sensitive to incremental changes in $\sigma_{MBL}$.
**4.5 Aerosol-cloud interactions under different regimes**

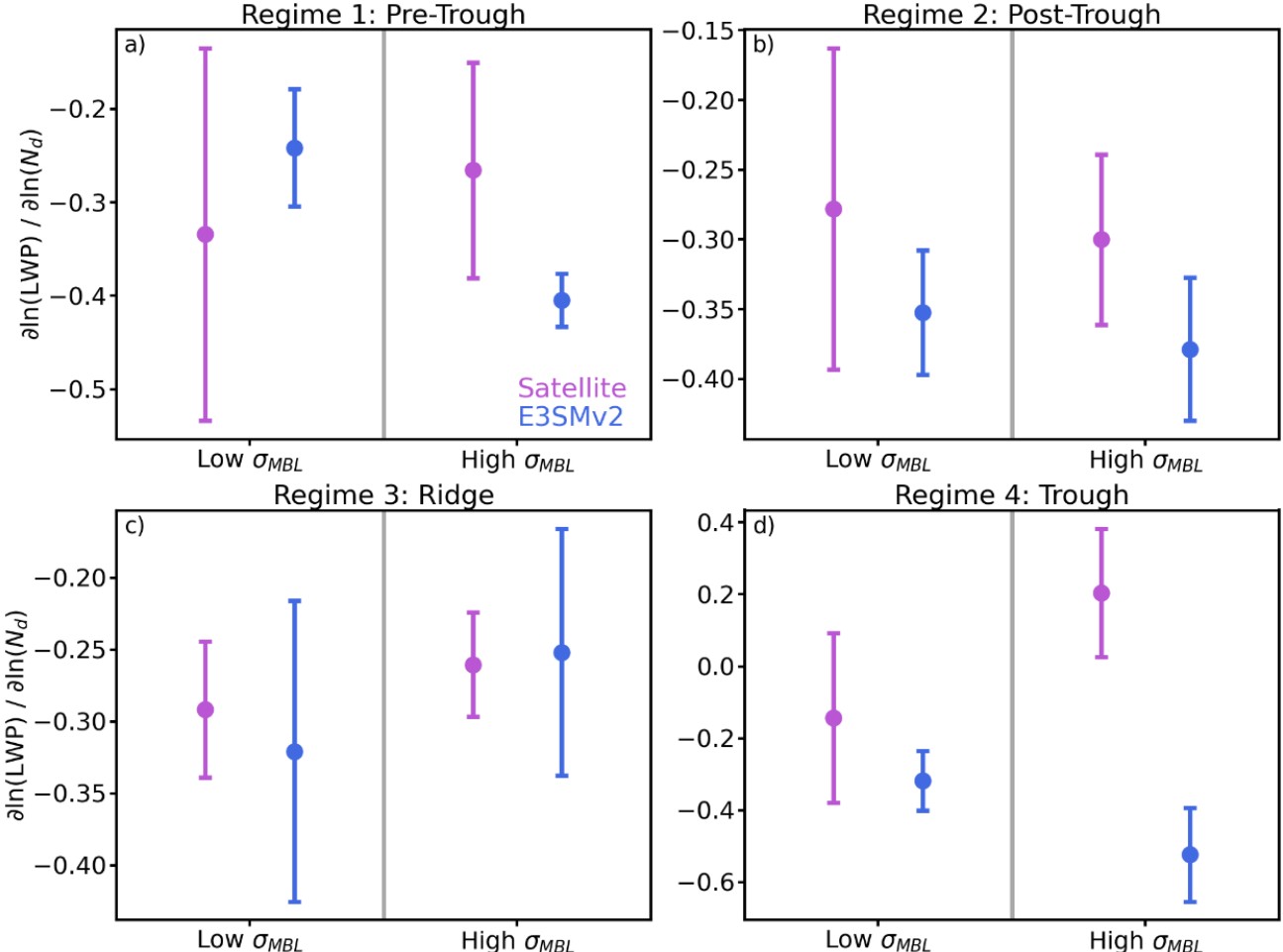

**Figure 9.** LWP adjustment due to $N_d$ under low and high $\sigma_{MBL}$ categories, separated by the median $\sigma_{MBL}$ values (gray line) from the aggregate satellite (purple) and E3SMv2 (blue) dataset. For a) Regime 1; b) Regime 2; c) Regime 3; d) Regime 4. Error bars denote standard errors of fitted slopes in each category.

In order to further illustrate the impact of aerosols on the behavior of the LWP–$N_d$ relationship,
both satellite and E3SMv2 data are grouped into lower and higher half $\sigma_{MBL}$ categories, defined by the
pooled median of the combined satellite and E3SM samples (0.594),  and this single threshold is applied
to all regimes and both datasets to ensures an identical conditioning (Fig. 9).
In the Pre-Trough regime (Fig. 9a), clouds exhibit low tops and moderate precipitation, featuring
a regime where precipitation suppression competes with entrainment drying. Satellite retrievals show a
weaker negative LWP–$N_d$ sensitivity under high aerosol loading (i.e., LWP declines less steeply with
$N_d$), whereas E3SMv2 simulates a steeper negative slope. The satellites may observe a weaker decline in

LWP with increasing $N_d$ because moderate precipitation in this regime allows sub-cloud drizzle evaporation to moisten the boundary layer, which weakens the in-cloud humidity gradient and reduces the entrainment efficiency, thereby buffering LWP losses (Wang et al., 2010; Chen et al., 2011; Jia et al., 2022). In the Ridge regime (Regime 3; Fig. 9c), characterized by shallow clouds and minimal precipitation, both satellite retrievals and E3SMv2 simulations yield a relatively weak (less negative) LWP–$N_d$ sensitivity under high aerosol conditions. At higher aerosol loadings, increased $N_d$ suppresses precipitation more effectively in these shallow clouds, stabilizing LWP by limiting drizzle loss. This mechanistic alignment between observations and E3SMv2 may explain their convergence toward a weaker negative slope (Quaas et al., 2020; Jia et al., 2022).

Post-Trough clouds (Regime 2; Fig. 9b), with greater cloud-top heights and moderate precipitation, exhibit a steeper negative LWP–$N_d$ sensitivity in both observations and E3SMv2. The greater depth of Post-Trough clouds amplifies the vertical moisture gradient between the cloud layer and the overlying dry free troposphere. In this scenario, the primary mechanism reducing LWP is enhanced entrainment-driven evaporation. Once droplets are smaller (high $N_d$), they evaporate faster at cloud top, enhancing evaporative cooling. This strengthens entrainment through buoyancy reversal, creating positive feedback that accelerates LWP loss (Gryspeerdt et al., 2019). Although precipitation suppression can still play a role, the net result in a deeper cloud is often dominated by entrainment drying rather than by the retention of liquid water. Consequently, as aerosol loading increases, the slope of LWP–$N_d$ becomes more negative, aligning with findings that deeper clouds, with stronger inversions, exhibit steeper negative slopes in LWP–$N_d$ (Zhang and Feingold, 2023).

The Trough regime (Fig. 9d), marked by deep, precipitating clouds and unstable conditions, highlights a key model-observation discrepancy: satellites detect a weak or even positive LWP–$N_d$ relationship at high aerosol loadings, while E3SMv2 simulates a steeply negative slope. It could be possible that, under the moist and unstable environment, the increasing $N_d$ provides more surface areas for water vapor condensation and hence offsets the entrainment drying loss (Gryspeerdt et al., 2019). Also, under this regime with relatively low $N_d$, the increasing evaporation is favorable for more latent heat release, allowing the cloud to be invigorated and expand in vertical extent, hence increasing the LWP (Altaratz et al., 2014). Moreover, satellite retrievals in this regime may be also biased by vertical cloud inhomogeneity and drizzle contamination, which could artificially inflate LWP estimates in high-$N_d$ conditions (Zhang et al., 2022). Meanwhile, E3SMv2 relies on parameterizations for turbulence and entrainment that are calibrated for shallow stratocumulus and may not fully capture the intermittency of entrainment in thicker, more precipitating cloud regimes (Gettelman and Morrison, 2015; Wang et al., 2023; Tang et al., 2023). This may cause the model to amplify entrainment drying relative to what might

be observed, thereby producing a more strongly negative LWP–$N_d$ slope than indicated by satellites.
Ultimately, these contrasting signals reflect both retrieval limitations in complex cloud systems and the
model's sensitivity to microphysical closure assumptions (Christensen et al., 2023).

Such model-satellite discrepancies are further confirmed in the bulk indirect susceptibility, which

quantifies the integrated response of LWP to changes in marine boundary layer aerosol extinction. As
shown in Table 4, satellite-derived susceptibilities range from –0.070 to –0.015, while E3SMv2
simulations consistently yield larger negative values (from –0.138 to –0.098), especially for Regime 2
and 3, which feature more stratiform-like clouds. That is, for a given increase in aerosol loading, E3SMv2
predicts a stronger LWP reduction than satellite observations. This systematic overestimation by
E3SMv2 indicates that the model may be overly sensitive to aerosol perturbations, translating into an
exaggerated indirect effect on LWP. Potential causes for this discrepancy include overly aggressive
drizzle suppression in the MG2 microphysics scheme, which may retain excess LWP at low $N_d$ but
prematurely truncate water accumulation as $N_d$ increases. In addition, limitations in representing subgrid-
scale turbulent mixing and entrainment could further contribute to the observed biases. While E3SMv2
captures the qualitative trends of aerosol–cloud interactions, the quantitative discrepancies highlight the
need for further refinement in aerosol activation and cloud microphysical parameterizations, as well as
improved process-level representations of drizzle processes and entrainment mixing. Addressing these
issues is essential for enhancing the model ability to simulate the indirect effects of aerosols on cloud
properties in the ENA region.

**Table 4. Indirect susceptibility of LWP to Aerosol for Satellite and E3SMv2**

| | Regime 1 | Regime 2 | Regime 3 | Regime 4 |
|---|---|---|---|---|
| $\dfrac{\partial ln(LWP)}{\partial ln(N_d)}$ | | | | |
| Satellite[*] | -0.164 ± 0.019 | -0.191 ± 0.016 | -0.176 ± 0.007 | -0.028 ± 0.046 |
| E3SMv2[*] | -0.314 ± 0.011 | -0.262 ± 0.013 | -0.356 ± 0.007 | -0.167 ± 0.029 |
| $\dfrac{\partial ln(N_d)}{\partial ln(\sigma_{MBL})}$ | | | | |
| Satellite[*] | 0.411 ± 0.056 | 0.364 ± 0.061 | 0.288 ± 0.032 | 0.518 ± 0.188 |
| E3SMv2[*] | 0.430 ± 0.025 | 0.527 ± 0.024 | 0.275 ± 0.015 | 0.616 ± 0.062 |
| $\dfrac{\partial ln(LWP)}{\partial ln(N_d)} * \dfrac{\partial ln(N_d)}{\partial ln(\sigma_{MBL})}$ | | | | |
| Satellite[†] | -0.067 ± 0.012 | -0.070 ± 0.013 | -0.051 ± 0.006 | -0.015 ± 0.024 |
| E3SMv2[†] | -0.135 ± 0.009 | -0.138 ± 0.009 | -0.098 ± 0.006 | -0.104 ± 0.021 |

[*]Values are slopes from ordinary least squares fits in log–log space. The '±' entries are standard errors.

[†]The errors for the product is obtained by propagating the standard errors of the component ($\partial lnLWP/\partial lnN_d$ and $\partial lnN_d/\partial ln\sigma_{MBL}$) slopes (multiplicative propagation).

**5. Summary and Discussions**
This study investigated aerosol-cloud interaction processes over the ENA by assessing satellite
retrievals and simulations from E3SMv2. Using newly developed CALIPSO-derived vertical-resolved
aerosol extinction and the collocated MODIS cloud properties, along with model output from a 1° nudged
E3SMv2 simulation over a ~10°×10° domain from 2006 to 2014. Leveraging a novel regime-based
evaluation framework that places E3SMv2 and satellite observations side-by-side within the clustered
synoptic regimes, the variations of marine low-cloud and aerosol properties, and the cloud responses to
aerosols are examined.

Our analysis reveals distinct seasonal variations in cloud and aerosol properties, with satellites and E3SMv2 exhibit higher $N_d$ with lower LWP in warm seasons and lower $N_d$ with higher LWP in cold seasons. E3SMv2 consistently underestimates $N_d$ while producing more comparable LWP, in line with prior work. In general, satellite retrievals and E3SMv2 simulations capture the qualitative trends where higher $N_d$ under increased aerosol loading and a characteristic inverted-V relationship between LWP and $N_d$. However, while the LWP response to $N_d$ in the satellite dataset is primarily attributed to precipitation suppression and entrainment and evaporative processes, the model simulations generate a similar but more dramatic shape due to deficiencies in representing these processes within the model parameterizations. Furthermore, it is possible that satellite retrieval biases and sampling strategies may contribute to the observed inverted-V behavior in the LWP-$N_d$ relationship (Grosvenor et al., 2018; Arola et al., 2022; Gryspeerdt et al., 2022), though we addressed retrieval uncertainties to some extent by applying several data screenings documented in Painemal et al. (2020). Another possibility is that the inverted-V shape may reflects natural spatial variability that leads to both increases and decreases of $N_d$ with LWP (Goren et al., 2025).

To better isolate meteorological controls, we employ a deep-learning-based clustering method to partition the ENA meteorology into four distinct synoptic regimes. Namely, Pre-Trough (regime 1), Post-Trough (regime 2), Ridge (regime 3) and Trough (regime 4). This clustering approach captures spatial and temporal variability more effectively than traditional methods and reveals that meteorology strongly modulates aerosol-cloud interactions. The inverted-V LWP-$N_d$ relationship persists across regimes, with regime dependent differences between satellites and E3SMv2.

Regime 3 (Ridge) shows the strongest simulated inverted-V response among all four regimes, where LWP decreases at high $N_d$, driven by the stronger contribution of the entrainment-drying effect in a thinner, more stable cloud layer. In the pre-trough regime, the LWP decline with high $N_d$ is partially masked by the interplay between precipitation-stabilization and entrainment-mixing processes. In the Post-Trough regime, deeper clouds with intermediate precipitation display a more negative LWP-$N_d$ sensitivity due to enhanced entrainment-drying effect, a process the model appears to overestimate compared to observations. Conversely, the Trough regime, characterized by deep, precipitating clouds, features complex behavior where satellite retrievals yield a muted LWP response, likely because of precipitation-suppressing overwhelming the entrainment-drying effect on LWP. The model overpredicts the LWP responses across all regimes with different magnitudes. Both datasets show increasing LWP with $N_d$ at low $N_d$, but E3SMv2 peaks at a lower $N_d$, suggesting overly aggressive drizzle suppression and insufficient representation of subgrid moisture variability. Moreover, the E3SMv2 simulates steeper

declines of LWP with $N_d$ at higher values, indicating that the model parameterization induces more rapid entrainment-drying effects contributing to excessive LWP loss.

Both satellites and E3SMv2 show $N_d$ increasing with MBL aerosol extinction in every regime. However, model $N_d$ susceptibility varies more with meteorology than the satellite record. In the regimes with relatively shallow clouds, both data and model exhibit the lower $N_d$ susceptibility to aerosol. In contrast, E3SMv2 shows a markedly steeper $N_d$ response than observed in regimes with deeper and more precipitating clouds, likely due to an overestimation of aerosol emission and activation or underrepresentation of limiting processes such as insufficient drizzle scavenging. Such discrepancies suggest that the parameterizations in model aerosol activation schemes might be overly sensitive to environmentally controlled factors, which leads to a larger range of $ACI_N$.

The regime-wise LWP responses on $N_d$ are analyzed under low and high MBL aerosol extinction conditions. In shallower and less precipitating cloud conditions, such as the Ridge regime, both satellite retrievals and E3SMv2 simulations converge to a weaker (less negative) sensitivity under high aerosol loading. By contrast, in more precipitating and vertically extended regimes, E3SMv2 exhibits significantly stronger LWP depletions on $N_d$ with increasing aerosols. The E3SMv2 operates at relatively coarse horizontal and vertical resolutions, and the parameterized microphysics. Previous sensitivity studies indicate that changes in low clouds from E3SMv2 are noticeably controlled by CLUBB, and followed by MG2, tunings (Zhang et al., 2023). And the MG2 scheme, tends to overemphasize the entrainment-driven drying and droplet evaporation as it was calibrated for shallow stratus and stratocumulus conditions (Tang et al., 2024). As a result, the transition from shallow to deep cloud regimes, where natural processes evolve continuously, may not be adequately captured, leading to an exaggerated drying signal in the model. While independent analysis of the MG2 scheme show bias in warm rain processes, as it realizes the negative LWP pathway too rapidly and strongly (Zhou et al., 2025), implicating the turbulence-microphysics coupling as a persistent bias source that aligns with our regime-specific over-depletion of LWP at high $N_d$. Regarding model resolution, vertically resolved physics and concurrent horizontal and vertical refinement improve the representation of entrainment mixing processes and reduce stubborn stratocumulus biases (Lee et al., 2022; Bogenschutz et al., 2023). Hence, we cautiously attribute the potential E3SMv2 discrepancies versus satellite results to those simulated processes in the model, while acknowledging that they can be also the combined effects from multiple feedback and interplay between the model schemes.

It is noteworthy that previous ACI studies in a synoptic context have been largely cyclone-centric (e.g., McCoy et al., 2020; Lee et al., 2025). Our regime-stratified results are consistent with that literature: in cyclone-associated conditions (Pre-Trough, Trough) we see LWP increases or smaller decreases with

higher Nd, whereas in the anticyclonic conditions (Ridge) LWP decreases markedly with higher Nd, as expected in stable, dry high-pressure environments. Our clustering approach extends the synoptic pattern classification by providing a flexible, data-driven identification that captures the same physical contrasts as cyclone masks while explicitly considering the other two regimes (Ridge and Post-Trough). Therefore, our approach might be more general, and remaining applicable beyond the regions dominated by midlatitude storm tracks. Model behavior also parallels prior findings: E3SMv2's overly steep LWP reductions in Ridge conditions mirror the overestimation of LWP sensitivity outside cyclones reported by McCoy et al. In short, our aim is to develop a data-driven way to untangle meteorology from cloud responses without pre-specifying synoptic systems, and the learned regimes would naturally recover the traditional cyclone phases (pre-, post-trough, trough). And we also view this as transferable to other regions of the globe, including marine stratocumulus regions with weaker cyclone influence (e.g., the southeastern Atlantic).

In summary, our findings report the satellite-observed and model-simulated range of aerosol-cloud interaction indices from both seasonal and regime-based perspectives over the ENA. Moreover, the regime-based analysis demonstrates that the interplay between aerosol loading and cloud microphysics is highly sensitive to the prevailing meteorological conditions. While E3SMv2 reliably reproduces the overall trends in aerosol effects on stratiform clouds, its performance degrades in deeper, more dynamically complex regimes. Given uncertainties in the satellite observations, it is critical for future studies to integrate datasets from airborne, ground-based, and satellite platforms. This strategy would enable the quantification of errors as well as corroborating the results presented here. In terms of the feasibility of potential model improvements, we think that a feasible approach would be the fine-tuning of the microphysical parameterization, ideally constrained by high-resolution observational data from field campaigns such as ARM. This may reduce the persistent uncertainties in simulating aerosol-cloud interactions, particularly under the dynamic meteorological transitions typical of the ENA region. Furthermore, emulation from high-resolution modeling (e.g., LES) of cloud and rain microphysics processes can be used to replace the bulk microphysics scheme, which can contribute to better performance with manageable cost as shown in previous studies such as Gettelman et al. (2021). Increasing spatial resolution is also feasible in a regionally refined mesh, and increasing vertical resolution might follow, but both would noticeably increase computational cost, so trade-offs should be considered with caution. Lastly, the development of new schemes that bridge the gap between shallow and deep cloud regimes remains particularly challenging, as current large-scale model schemes still treat them separately.

Future research will focus on exploring the transferability of this regime-based analysis to other

global marine regions, assessing the scaling effects and exploring the process-level understanding of

aerosol-cloud interactions within models, and extending the investigation to include aerosol-cloud-

radiation interactions, thereby providing better constraints on effective radiative forcing. Such efforts are

essential to refine microphysical parameterizations and enhance the overall fidelity of climate models in

representing these critical processes.

**Code and Data availability**

The E3SMv2 nudged simulation output is available at: https://zenodo.org/records/15670340. The ERA5 reanalysis is available at: https://cds.climate.copernicus.eu/datasets/reanalysis-era5-pressure-levels?tab=overview. The original clustering model is available at: https://zenodo.org/records/14720991. The hyperparameter-tuned model and the collocated CALIPSO-MODIS dataset are available upon request.

**Author contributions**

The idea of this study was developed by XZ, YF, and DP. XZ performed the analyses and wrote the manuscript under the supervision of YF. MZ performed the nudged E3SM simulation. DP and ZL constructed the collocated CALIPSO-MODIS dataset. XZ, YF, DP, MZ, SX, ZL, RJ and BL participated in further scientific discussions and provided substantial comments and edits on the paper.

**Competing interests**

At least one of the (co-)authors is a member of the editorial board of Atmospheric Chemistry and Physics.

**Acknowledgement**

This research was funded by the CloudSat and CALIPSO Science Team Recompete Program under the Science Mission Directorate of NASA (NNH21ZDA001N-CCST). Y.F., M.Z., S.X., and R.J. would like to acknowledge the support of the Energy Exascale Earth System Model (E3SM) project, Y.F. also acknowledges the support of the Atmospheric System Research (ASR) program; both projects are funded by the U.S. Department of Energy (DOE), Office of Science, Office of Biological and Environmental

Research. The work at Argonne National Laboratory was supported by the U.S. DOE Office of Science under contract DE-AC02-06CH11357. Work at Lawrence Livermore National Laboratory was performed under the auspices of the US DOE by Lawrence Livermore National Laboratory under contract No. DE-AC52-07NA27344. We acknowledge the computing resources provided on Improv, a high-performance computing cluster operated by the Laboratory Computing Resource Center at Argonne National Laboratory.

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
