# Peer review of "Regime-based Aerosol-Cloud Interactions from CALIPSO-MODIS and the Energy Exascale"

_EGUsphere, 2025_

## Referee Comment (RC1)

**Review of "Regime-based Aerosol-Cloud Interactions from CALIPSO-MODIS and the Energy Exascale Earth System Model version 2 (E3SMv2) over the Eastern North Atlantic"**

**Summary**

This work studies aerosol-cloud interactions in the Eastern North Atlantic region, comparing satellite retrievals to a GCM model output, for a large set of days, to elucidate the relationship between micro- and macrophysical cloud properties: cloud droplet number concentration, liquid water path, and boundary-layer extinction coefficient. They analyze the relationships between these variables seasonally, comparing simulations to satellite data, and the main novelty of the study is that they also analyze the behavior for 4 meteorological regimes that are found using clustering techniques on ERA5 reanalysis data. This regime clustering gives new insights by separating natural covariability and clarifying one of the relationships. The paper is well written, and the discussion is very detailed and provides a full understanding of the studied system and its physical processes. I mostly have minor comments regarding some methods, and about how to better summarize and provide ideas to modelers based on their discussion.

**Minor comments**

- I suggest highlighting the novelties of the paper in the abstract, introduction, and summary. In particular, I am not sure if the novelty is only the analysis based on regimes, or if the seasonal analysis is also novel? Or is this particular model and satellite product comparison new?

- The discussion in every Section is very thorough, but many hypotheses point to modeling biases or ideas for model improvements, which are not the main scientific contribution of this work. It would be nice to assess if these hypotheses are true by confirming some diagnostics on the resulting model parameters. Another thing that could be done is to order these recommendations and try to assess which model improvements are more likely or feasible.

**Line by line comments**

- L17 Clustering was performed on satellite or simulation data? Or both?

- L18 Maybe explain the 4 regimes before they start appearing

- L161 Are there comparisons for other cloud types?

- L195 What is the value of that coarse vertical resolution?

- L213 Time formatting: Should it be 1 p.m or 13:00?

- L214 Was the date also a variable?

- L226 So the DEC was used after optimizing the k-means clustering? Or was it also tested for different k values?

- L246 "followed by fall", "lowest during winter"?

- L278 Is this index computed from the data? Is it a fit with confidence interval?

- L355-359 This sentence is a bit confusing

- Fig. 5: Composites mean that these are based on the mean values of each cluster? Or are these the centroids?

- L401 Details were already given in the previous Section

- L407 Is there a reason why the regime order does not follow the expected trough-ridge transition?

- L426 I think it is important to report the number of events and percentage for each regime in the main manuscript, for statistical significance. Now that I see the supplementary information, maybe it is worth cautioning the readers that regime 4 had the lowest amount of information

- L442 "are listed"

- Fig. 9: The median sigma values were selected for each regime or for the entire dataset?

- L720 I suggest mentioning the four regimes

---

## Author Comment (AC1)

**Responses to Reviewer 1**

Dear Reviewer, we appreciate your time and effort in acknowledging and thoroughly reviewing our manuscript. We are truly grateful for your constructive comments and insightful suggestions, which encourage and help us to improve the manuscript. We have revised the manuscript carefully based on your comments.

In the responses below, your comments are provided in black text and our responses are provided in blue text.

This work studies aerosol-cloud interactions in the Eastern North Atlantic region, comparing satellite retrievals to a GCM model output, for a large set of days, to elucidate the relationship between micro- and macrophysical cloud properties: cloud droplet number concentration, liquid water path, and boundary-layer extinction coefficient. They analyze the relationships between these variables seasonally, comparing simulations to satellite data, and the main novelty of the study is that they also analyze the behavior for 4 meteorological regimes that are found using clustering techniques on ERA5 reanalysis data. This regime clustering gives new insights by separating natural covariability and clarifying one of the relationships. The paper is well written, and the discussion is very detailed and provides a full understanding of the studied system and its physical processes. I mostly have minor comments regarding some methods, and about how to better summarize and provide ideas to modelers based on their discussion.

We sincerely appreciate your thoughtful and constructive feedback. All comments have been carefully considered, and the manuscript has been revised accordingly.

**Minor comments**

• I suggest highlighting the novelties of the paper in the abstract, introduction, and summary. In particular, I am not sure if the novelty is only the analysis based on regimes, or if the seasonal analysis is also novel? Or is this particular model and satellite product comparison new?

The primary novelty is the regime-based evaluation framework that places E3SMv2 and satellite observations side-by-side within the same synoptic regimes, allowing like-for-like attribution of ACI behavior. A second novelty is our use of a new vertically resolved aerosol-extinction product to diagnose free-tropospheric versus boundary-layer influences on the LWP- $N_d$  relationship. The seasonal analysis is included chiefly to reassure and reconfirm prior findings and to show that the regime-based conclusions are robust across seasons.

We have revised the abstract, introduction, and summary to state these contributions explicitly.

• The discussion in every Section is very thorough, but many hypotheses point to modeling biases or ideas for model improvements, which are not the main scientific contribution of this work. It would be nice to assess if these hypotheses are true by confirming some diagnostics on the resulting model parameters. Another thing that could be done is to order these recommendations and try to assess which model improvements are more likely or feasible.

We appreciate the suggestion to verify our hypotheses with targeted parameter diagnostics and to prioritize feasible improvements. However, our study's main contribution is to provide a novel regime-based evaluation which helps narrow down the conditions where the model uncertainties in ACI are the largest and further identify the processes associated with those specific regimes. Analyses of comprehensive parameter perturbations require a set of paired simulations that are beyond the scope of this work. Instead, we carefully grounded our attributions to published model sensitivity and resolution studies. Previous sensitivity studies indicate that changes in low clouds in E3SMv2 are primarily controlled by CLUBB, followed by MG2 tunings (Zhang et al., 2023). Independent analysis of the MG2 scheme show biases in warm-rain processes and that it realizes the negative LWP pathway too rapidly and strongly (Zhou et al., 2025), implicating the turbulence-microphysics coupling as a persistent bias source that aligns with our regime-specific over-depletion of LWP at high  $N_d$ . Moreover, model resolution studies found that vertically resolved physics and concurrent horizontal and vertical refinement improve the representation of entrainment mixing processes and reduce stubborn stratocumulus biases (Lee et al., 2022; Bogenschutz et al., 2023). Hence, we cautiously attribute the potential E3SMv2 discrepancies

versus satellite results to those simulated processes in the model, while acknowledging that they can also be the combined effects of multiple feedbacks and interplay among the model schemes.

In terms of the feasibility of potential model improvements, we think that a feasible approach would be the fine-tuning of the microphysical parameterization, ideally constrained by high-resolution observational data from field campaigns such as ARM. This may reduce the persistent uncertainties in simulating aerosol-cloud interactions, particularly under the dynamic meteorological transitions typical of the ENA region. Furthermore, emulation from high-resolution modeling (e.g., LES) of cloud and rain microphysics processes can be used to replace the bulk microphysics scheme, which can contribute to better performance with manageable cost as shown in previous studies such as Gettelman et al. (2021). Increasing spatial resolution is also feasible in a regionally refined mesh, and increasing vertical resolution might follow, but both would noticeably increase computational cost, so trade-offs should be considered with caution. Lastly, the development of new schemes that bridge the gap between shallow and deep cloud regimes remains particularly challenging, as current large-scale model schemes still treat them separately.

We have added the above discussions in the revised Section 5.

**Line by line comments**

• L17 Clustering was performed on satellite or simulation data? Or both?

The clustering is applied on the ERA5 reanalysis, then the satellite and model data are aggregated based on the clustered regimes.

• L18 Maybe explain the 4 regimes before they start appearing

We have revised this statement to 'We then partition ENA meteorology into four synoptic regimes (Pre-Trough, Post-Trough, Ridge, Trough) via a deep-learning clustering of ERA5 reanalysis fields'.

**• L161 Are there comparisons for other cloud types?**

Yes, Gryspeerdt et al. (2022) explicitly compares satellite-retrieved  $N_d$  with in-situ aircraft data across multiple cloud regimes. They find high fidelity in marine stratocumulus and lower correlations in more challenging convective situations. And this is precisely why, in our study, we prioritize low-level liquid clouds, where the satellite retrieval is best-validated and most defensible for model-satellite ACI evaluation.

We have revised the statement to '...previous studies have shown that the  $N_d$  compares well with measurements from 11 aircraft campaigns, demonstrating a decent correlation when sampling the marine stratocumulus clouds, with  $r^2$  values of 0.5~0.8 (Gryspeerdt et al., 2022). Therefore, to minimize known retrieval uncertainties, we focus on low-level liquid clouds where satellite  $N_d$  shows the strongest aircraft agreement and typical normalized root mean squared deviation of ~30-50 % (Gryspeerdt et al., 2022).'

**• L195 What is the value of that coarse vertical resolution?**

Our E3SMv2 simulation employs the standard  $\sim$ 72-layer atmosphere, giving a near-surface vertical resolution of roughly 50–100 m, gradually coarsening to  $\sim$ 200–300 m per layer near cloud top through the free troposphere.

For a rough estimate, we have revised the statement to 'Given the coarse vertical resolution of E3SM near the cloud top ( $\sim$ 200-300 m), ...'

**• L213 Time formatting: Should it be 1 p.m or 13:00?**

We have changed the occasion to '13:00 LT (1 p.m. local time)' for more consistent formatting.

**• L214 Was the date also a variable?**

We did not include calendar date (or month-of-year) as an input feature; the model ingests only Z500, SLP, and 10-m winds, using temporal ordering (not absolute time) for the LSTM. Since

adding date would impose a seasonal prior that can bias the clustering toward calendar timing rather than physical flow patterns.

• L226 So the DEC was used after optimizing the k-means clustering? Or was it also tested for different k values?

Yes, the DEC was used after optimizing the k-means clustering. We first determined the optimal number of clusters k, and then ran DEC with that fixed k (4 in this study). DEC was initialized with the k-means centroids and then optimized the KL-divergence (KL) clustering loss between the encoder's soft assignments and a sharpened target distribution, with periodic centroid updates, and the k remained the k-means-optimized value throughout.

**We have clarified the methodology as follows:**

"...To further refine the clustering assignments, we then ran DEC with that fixed cluster number of four, as determined with K-means optimization. DEC was initialized by the K-means centroids and optimized the KL-divergence clustering loss (between soft assignments and a sharpened target distribution) with periodic centroid updates..."

• L246 "followed by fall", "lowest during winter"?

Thanks for the correction. We have revised it to 'followed by fall (SON, 73.36 cm-3), and the lowest during winter (DJF, 60.37 cm-3)'

• L278 Is this index computed from the data? Is it a fit with confidence interval?

Yes. The adjustment index  $\mathcal{L}_0$  is computed directly from the data for satellites and for E3SMv2 separately.

The reported " $\pm$ " values are the standard error (SE) of the slope of ordinary least squares fit in log-log space (scipy.stats.linregress). A 95% confidence interval can be reported as  $\mathcal{L}_0 \pm 1.96 \times SE$ . Here our main text currently shows the slope  $\pm$  SE.

**We have clarified the methodology as follows:**

'We compute  $\mathcal{L}_0$  as the slope of an ordinary least squares fit in log-log space between  $N_d$  and LWP. Hence, the  $\mathcal{L}_0$  derived from satellite observations and model simulations is -0.192  $\pm$  0.006 and -0.375  $\pm$  0.005, respectively. The ' $\pm$ ' values reported are the standard errors of the slope (SE) from that fit (equivalently, 95% confidence level CI = slope  $\pm$  1.96\*SE, under standard linear-regression assumptions).'

**• L355-359 This sentence is a bit confusing**

We have revised the discussion for better clarity:

'Aircraft in situ measurements near cloud base provide the most physically robust ACI assessment (Gupta et al., 2021; Zheng et al., 2024). However, it is challenging to do that with satellite data and model outputs, because satellite remote sensing like CALIOP cannot reliably determine cloud-base height, and the model's coarse vertical resolution makes it difficult to collocate the model cloud-base with CALIOP layers. Hence, those factors necessitate the use of the mean aerosol properties within the below-cloud-top MBL in the present study'

- Fig. 5: Composites mean that these are based on the mean values of each cluster? Or are these the centroids?
- & L401 Details were already given in the previous Section

For each regime we average the ERA5 fields over all dates classified into that regime to form the maps shown (Z500 with winds, SLP with winds, and LTS). Thus each panel represents the mean state of all members in that cluster.

**Hence, we have revised description of Figure 5 for better clarity below:**

'As detailed earlier, the CNN-LSTM-DEC clustering of 3,286 daily ERA5 states results in the identification of four distinct synoptic - scale regimes (Figure 5). Namely, Pre-Trough (regime 1), Post-Trough (regime 2), Ridge (regime 3) and Trough (regime 4). For each regime, composites

were computed as the arithmetic mean of the corresponding ERA5 fields across all time steps assigned to that regime.'

• L407 Is there a reason why the regime order does not follow the expected trough-ridge transition?

We decided to pair Ridge with Trough in figures and discussion to create a clear side-by-side contrast, and Pre-Trough with Post-Trough because they have comparable sample sizes and bracket the Trough disturbance. And the regime labels are permutation-invariant outputs of the unsupervised clustering. We order them for readability rather than chronology, so numbering should not be interpreted as a trough–ridge time sequence.

• L426 I think it is important to report the number of events and percentage for each regime in the main manuscript, for statistical significance. Now that I see the supplementary information, maybe it is worth cautioning the readers that regime 4 had the lowest amount of information

Thanks for the suggestion, and Reviewer 2 also raised similar comments.

Hence, we have moved Table 1 to the main text. And added the following statement: 'Note that among the four regimes, Regime 4 is the least frequent (3.4%) and is largely confined to the colder seasons (winter and spring), confirming the findings from previous studies'

• L442 "are listed"

Thanks, correction has been made in the revised manuscript.

• Fig. 9: The median sigma values were selected for each regime or for the entire dataset?

We used a single median threshold for  $\sigma_{MBL}$  computed from the pooled satellite and E3SM dataset rather than regime- or dataset-specific medians. This ensures an identical conditioning for both data sources and all regimes, avoiding different bin edges that could confound interpretation.

**We have clarified that as follows:**

'In order to further illustrate the impact of aerosols on the behavior of the LWP– $N_d$  relationship, both satellite and E3SMv2 data are grouped into lower and higher half  $\sigma_{MBL}$  categories, defined by the pooled median of the combined satellite and E3SM samples (0.594), and this single threshold is applied to all regimes and both datasets to ensures an identical conditioning (Fig. 9).'

• L720 I suggest mentioning the four regimes

The four regimes are now mentioned in the beginning of this paragraph.

---

## Author Comment (AC2)

**Responses to Reviewer 2**

Dear Reviewer, we appreciate your time and effort in acknowledging and thoroughly reviewing our manuscript. We are truly grateful for your constructive comments and insightful suggestions, which encourage and help us to improve the manuscript. We have revised the manuscript carefully based on your comments.

In the responses below, your comments are provided in black text and our responses are provided in blue text.

Review of "Regime-based Aerosol-Cloud Interactions from CALIPSO-MODIS and the Energy Exascale 2 Earth System Model version 2 (E3SMv2) over the Eastern North Atlantic" by Zheng et al.

In this work, the authors investigate aerosol-cloud interactions (ACI) in the Eastern North Atlantic (ENA) with satellite observations and nudged Energy Exascale Earth System Model version 2 (E3SMv2) simulations. In particular, the authors examine differences in liquid water path (LWP), droplet number concentration (Nd), and their covariance between observations and simulations. They find that, in general, there are systematic seasonal discrepancies between E3SMv2 and satellite observations of LWP and Nd that line up with prior studies. They also find the presence of the "inverted-V" in the models and observations, with a more pronounced V shape in the model output. To investigate the effect of ENA meteorology on these results, the authors employ a machine learning method to partition the data into 4 regimes based on synoptic conditions: pretrough, post-trough, ridge, and trough. This more-targeted analysis reveals additional, interesting insights into the differences between the model and the observations and allow for more specific inferences on the importance of meteorology on ACI processes.

The paper is well-organized and detailed, showcasing valuable results that are interesting on their own merit and motivate exciting future research. The reanalysis-clustering method for regime analysis in particular I thought was a strong result with broad-reaching applications. ENA is strongly governed by transient synoptic weather systems, and I felt this methodology provided an interesting alternative to the usual methods of regime classification. While I think this is overall a

strong paper, I do have some comments that I feel should be addressed before publication. These are detailed below. Good work!

We sincerely appreciate your thoughtful and constructive feedback. All comments have been carefully considered, and the manuscript has been revised accordingly.

**General comments**

One key problem I have with this paper is that I think it "buries the lede" in terms of its subject matter. In the abstract, the regime names seem to come out of nowhere, and the title gives no indication that synoptic regimes are a key piece of subject matter for this work. Synoptic systems as an important lens through which we should be viewing this data aren't suggested until well into the paper (end of section 2) and aren't truly discussed until section 4. While I understand the desire to maintain a clear narrative, I think highlighting this portion of the analysis more clearly in the abstract and motivating it more directly in the introduction would help strongly with readability.

Thanks for the suggestion. We have carefully revised the abstract, introduction, and the narrative in Section 3 to better reflect the synoptic regime-based framework of the analyses.

**The new abstract now reads:**

'Abstract. This study investigates aerosol-cloud interactions in marine boundary layer (MBL) clouds using an advanced deep-learning-driven synoptic-regime-based framework, combining satellite data (CALIPSO vertically resolved aerosol extinction and MODIS cloud properties) with  $1^{\circ}$  nudged Energy Exascale Earth System Model version 2 (E3SMv2) simulation over the Eastern North Atlantic (ENA; ~ $10^{\circ}$ × $10^{\circ}$ , 2006-2014). The E3SMv2 captures observed seasonal variations in cloud droplet number concentrations ( $N_d$ ) and liquid water path (LWP), though it systematically underestimates  $N_d$ . We then partition ENA meteorology into four synoptic regimes (Pre-Trough, Post-Trough, Ridge, Trough) via a deep-learning clustering of ERA5 reanalysis fields, enabling regime-dependent aerosol-cloud interactions analyses. Both satellite and E3SMv2 exhibit an inverted-V LWP– $N_d$  relationship. In Post-Trough and Ridge regimes, the satellite shows stronger negative LWP– $N_d$  sensitivities than in Pre-Trough regime. The Trough regime displays a muted

satellite LWP response. In comparison, the model predicts more exaggerated LWP responses across regimes, with LWP increasing too quickly at low  $N_d$  and decreasing more sharply at high  $N_d$ , especially in Pre-Trough and Trough regimes. These exaggerated model LWP sensitivities may stem from uncertainties in representing drizzle processes, entrainment, and turbulent mixing. As for  $N_d$  susceptibility to aerosols,  $N_d$  increases with MBL aerosol extinction in both datasets, but the simulated aerosol-cloud interactions appear oversensitive to meteorological conditions. Overall, E3SMv2 better captures aerosol effects under regimes that favor stratiform clouds (Post-Trough, Ridge), but performance deteriorates for regimes with deeper, dynamically complex clouds (Trough), highlighting the need for improved representations of those cloud processes in climate models.'

**And we have highlighted the importance of synoptic regimes in assessing the ACI in the introduction:**

'Moreover, synoptic systems organize boundary-layer clouds on multi-day timescales and strongly modulate aerosol-cloud-precipitation pathways (Mechem et al., 2018; Lee et al., 2025). Therefore, quantifying the untangled aerosol-cloud sensitivities require conditioning on the synoptic environment. For example, Zhang et al. (2022) found that the relationship between LWP and  $N_d$  is not only sensitive to aerosol loading but also modulated by the underlying meteorological conditions. And McCoy et al. (2020) used a cyclone compositing approach to demonstrate that aerosol-cloud interactions (e.g., the sign of LWP change with  $N_d$ ) can differ inside vs. outside midlatitude cyclones. These considerations motivate our use of an objective synoptic-regime classification to control meteorology when evaluating the synoptic-regime-dependent ACI.'

In line with my previous comment, I feel the paper is a little lacking in terms of background on synoptic regime analysis with respect to ACI. While the authors do mention McCoy et al., 2020, which is important background for this work, it is done in the somewhat vague context of "atmospheric regimes" (particularly when "cloud regimes" and "meteorology regimes" are specified in the next few paragraphs). I think, generally, prior authors examining ACI in a synoptic

meteorology context have taken a more cyclone-specific approach (as in, compositing around low-pressure centers), as opposed to the trough/ridge classification approach. Differentiating synoptic regimes from more general "atmospheric" regimes here would help with seeding this idea early on. Also, I think adding some additional analysis contrasting these results with the prior literature in this area (e.g., McCoy et al., 2020) would add some critical context to the results/conclusions of this paper. If the authors feel that the analysis presented is too novel to be usefully compared to prior analyses of ACI in synoptic scale contexts, then that needs to be defended more thoroughly in the manuscript.

We appreciate the reviewer's feedback and agree that referencing cyclone-based ACI literature strengthens our work. Our goal is to introduce a data-driven methodology to untangle meteorology from cloud responses, allowing the machine-learning framework to decide what aspects of the atmospheric pattern matter, without relying on pre-identifying specific synoptic systems (e.g., cyclones). This approach performs well: the clustered patterns naturally encompass the traditional cyclone separation, including pre-trough, post-trough, and trough phases. We also view this as transferable to other regions of the globe, including marine stratocumulus areas with weaker cyclone influence (e.g., the southeastern Atlantic).

**That said, we have made several revisions to acknowledge and compare with prior work:**

We have added references to McCoy et al. (2020) and related studies in the Introduction to highlight the importance of synoptic regime in aerosol-cloud interactions, as stated in above response. We acknowledge prior evidence that meteorological context influences ACI, thereby motivating our use of a synoptic clustering framework. This addition makes it clear to readers that our approach is built on the foundation of such findings, as we are extending the idea of regime-dependent ACI that others have established.

In Section 5 of the revised manuscript, we explicitly contrast our findings with McCoy et al. (2020). They reported LWP increases with aerosol in cyclonic regimes and little or opposite effect outside cyclones; we see the same pattern in satellite observations, enhanced LWP at higher  $N_d$  in the Pre-Trough (cyclone) regime and suppressed LWP in the Ridge (non-cyclone) regime. We also note

that our model evaluation aligns with McCoy's diagnosis of climate model bias: just as UM GA7.1 overestimated LWP response outside cyclones, E3SMv2 tends to over-respond in the Ridge regime. We qualitative argue that our study's conclusions support and extend the conclusions of cyclone-focused studies, thereby firmly embedding our contributions in the context of existing literature, while offering a more flexible, generalizable pathway via objective synoptic clustering.

**The added discussion now reads:**

'It is noteworthy that previous ACI studies in a synoptic context have been largely cyclone-centric (e.g., McCoy et al., 2020; Lee et al., 2025). Our regime-stratified results are consistent with that literature: in cyclone-associated conditions (Pre-Trough, Trough) we see LWP increases or smaller decreases with higher Nd, whereas in the anticyclonic conditions (Ridge) LWP decreases markedly with higher Nd, as expected in stable, dry high-pressure environments. Our clustering approach extends the synoptic pattern classification by providing a flexible, data-driven identification that captures the same physical contrasts as cyclone masks while explicitly considering the other two regimes (Ridge and Post-Trough). Therefore, our approach might be more general, and remaining applicable beyond the regions dominated by midlatitude storm tracks. Model behavior also parallels prior findings: E3SMv2's overly steep LWP reductions in Ridge conditions mirror the overestimation of LWP sensitivity outside cyclones reported by McCoy et al. In short, our aim is to develop a data-driven way to untangle meteorology from cloud responses without prespecifying synoptic systems, and the learned regimes would naturally recover the traditional cyclone phases (pre-, post-trough, trough). And we also view this as transferable to other regions of the globe, including marine stratocumulus regions with weaker cyclone influence (e.g., the southeastern Atlantic).'

Throughout the manuscript, there are many figures with errorbars. It is unclear to me outside of figures 4 and 7) what precisely the ranges on these errorbars represent. In a note on Table 3, the range values are described as propagated uncertainties of variables used in the covariance. Is that the case for the bounds of the errorbars in the remainder of the figures? Or are they representing

other statistics of the distribution (e.g. standard deviation like Figs. 4, 7)? More thorough descriptions of these errorbars in their respective figure labels are necessary for reader understanding. Additionally, I think some more detail on the sources of uncertainty (as mentioned in Table 3) would be helpful for overall understanding. Specific uncertainty values are mentioned once in the paper on L139 for MODIS-retrieved LWP. It is clear, however, that other sources of uncertainty – from satellite observations and from simulations – are considered for calculating the bounds in Table 3 (and perhaps elsewhere). To the authors' credit, the uncertainties inherent to the utilized satellite retrievals (particularly with respect to their analysis of the trough regime) are discussed in the paper. But to contextualize the results, more thoroughly and specifically describing the sources of uncertainty and how they've factored into the analysis is necessary.

Thank you for pointing this out. We would like to clarify that we use two different notions of "error" in the manuscript:

1. Error bars in Figs. 4, 7, 8

These are standard deviations (SD) from the binned statistics shown in each panel (i.e., spread of the data that fall in that bin), and they do not represent propagated analytical uncertainty.

2. "±" values in Table 3 (now Table 4 in revised manuscript) and for slope estimates

These are standard errors (SE) of fitted slopes for the  $\partial \ln(LWP)/\partial \ln(N_d)$  and  $\partial \ln(N_d)/\partial \ln(\sigma_{MBL})$ .

As for the

$$\frac{\partial ln(LWP)}{\partial ln(N_d)} * \frac{\partial ln(N_d)}{\partial ln(\sigma_{MBL})} \,,$$

The uncertainty SE is obtained by propagating the two slope SEs. Assuming independence,

$$\frac{Var(AB)}{(AB)^2} \approx \frac{SE(A)^2}{A^2} + \frac{SE(B)^2}{B^2}, hence, SE(AB) \approx |AB| \sqrt{\left(\frac{SE(A)}{A}\right)^2 + \left(\frac{SE(B)}{B}\right)^2}$$

Here  $A = \partial \ln(LWP)/\partial \ln(N_d)$  and  $B = \partial \ln(N_d)/\partial \ln(\sigma_{MBL})$ . We have revised the corresponding figure and table captions to state this.

**As for the satellite instrumental uncertainties, we have added the following discussion in Section 2:**

'Although the relative errors in  $N_d$  retrieval can be significant at the pixel scale (Grosvenor et al., 2018), previous studies have shown that the  $N_d$  compares well with measurements from 11 aircraft campaigns, demonstrating a decent correlation when sampling the marine stratocumulus clouds, with  $r^2$  values of 0.5~0.8 (Gryspeerdt et al., 2022). Therefore, to minimize known retrieval uncertainties, we focus on low-level liquid clouds where satellite  $N_d$  shows the strongest aircraft agreement and typical normalized root mean squared deviation of ~30-50 % (Gryspeerdt et al., 2022). Moreover, the aggregated collocation method significantly reduces the MODIS Aqua  $N_d$  bias (Painemal et al., 2020), resulting in a relationship between aerosol and cloud properties less affected by artifacts.'

**Furthermore, we assessed how satellite retrieval uncertainty could affect the LWP- $N_d$ relationship $\mathcal{L}_0$ :**

Since satellite (MODIS)  $N_d$  study report the report normalized RMS differences of ~30–50 % against aircraft data, we modeled  $N_d$  errors as multiplicative, mean-one lognormal noise with coefficient of variation (CV) 30–50 %. In log-log Ordinary Least Squares (OLS) fits, this is equivalent to additive noise in  $\ln N_d$ . With log-space error standard deviation is  $\sigma_{\epsilon} = \sqrt{\ln (1 + CV^2)}$  ( $\approx 0.29$  for 30 % and  $\approx 0.47$  for 50 %), and  $\mu = -\frac{1}{2}\sigma_{\epsilon}^2$ , the effect in log-space is:

$$lnN_{d,obs} = lnN_{d,true} + \eta, \ \eta \sim N(\mu,\sigma_{\epsilon}).$$

Hence, the OLS with noise in the predictor  $lnN_d$  would experience the regression-dilution effect and the expected slope would be attenuated as:

$$\mathbb{E}[\mathcal{L}_0] \approx \mathcal{L}_0 * \frac{Var(lnN_d)}{Var(lnN_d) + Var(\eta)},$$

which implies  $\sim 10-30$  % potential slope damping. Consistent with this expectation, a Monte-Carlo sensitivity test, multiplying the satellite  $N_d$  with lognormal noise (coefficient of variation  $\sim 30-50$ %) and refitting for 1000 times, attenuated the bulk satellite slope from -0.192 (baseline) to a median of -0.157 ( $\sim 18$  % attenuation), with a 95 % sensitivity band of [-0.170, -0.143]. However, this

does not alter the sign or the comparative result we report, but it does indicate that observed magnitudes should be interpreted with caution.

For E3SMv2  $N_d$  there is no instrument error, its uncertainty manifests as model bias and state-dependent spread. We therefore treat model  $N_d$  as as-simulated and discuss the biases in the results, rather than inserting an additional error term into slope SEs.

**We have added the above discussion in the revised Section 3:**

'To gauge how satellite retrieval uncertainty affects the LWP versus  $N_d$  relationship, we note that satellite  $N_d$  studies report normalized root-mean-square differences of about 30 to 50% relative to aircraft data as in Gryspeerdt et al. (2022). In log space a multiplicative  $N_d$  error is additive in  $\ln N_d$ , implying an expected slope damping of roughly 10 to 30%. Consistent with this expectation, a Monte Carlo test that multiplied satellite  $N_d$  by lognormal noise with coefficient of variation ~30-50% and refit the slope 1000 times reduced the bulk satellite slope from -0.192 to a median of -0.157, an attenuation of about 18%, with a 95 percent sensitivity band of -0.170 to -0.143. However, this does not alter the sign or the comparative result reported here, but it does indicate that observed magnitudes should be interpreted with caution.'

**Specific comments:**

L185: What ERA5 outputs are used for nudging the model, specifically? What is the relaxation time? More information as to how the model has been nudged is necessary.

We have added the nudging detail as below:

'In this study, EAMv2 was run at standard resolution (~110 km) with the meteorology nudged to ERA5. The model was nudged toward the ERA5 zonal (U) and meridional (V) wind and temperature fields using a relaxation time of 6 h. This nudging reduces errors in the simulated meteorology, facilitating the examination of aerosol and cloud properties.'

L189: You define  $\sigma EXT$  already on L129

Thanks, it has been corrected now.

References: Mechem et al., 2018, is cited throughout the paper, but seems to be absent from your list of references.

This reference is corrected listed now.

Table S1: I found myself frequently referencing Table S1 during my review of this paper and feel that it would be a useful inclusion in the main manuscript.

We have moved the Table S1 to main text.

---

## Author Comment (AC3)

**Responses to Reviewer 3**

Dear Reviewer, we appreciate your time and effort in acknowledging and thoroughly reviewing our manuscript. We are truly grateful for your constructive comments and insightful suggestions, which encourage and help us to improve the manuscript. We have revised the manuscript carefully based on your comments.

In the responses below, your comments are provided in black text and our responses are provided in blue text.

This work uses a regime-based approach to investigate aerosol-cloud interactions, specifically LWP and Nd, in warm marine clouds. The data set incorporates 2006 to 2014 measurements from CALIPSO for aerosol properties with MODIS for cloud properties in comparison to E3SMv2 simulations with machine learning to cluster four synoptic regimes in the ENA. Overall, they find that model relationships match observations qualitatively, but have more sensitive relationships likely due to model representation of cloud processes.

The paper reads very well with very few typos. The authors do a good job presenting necessary background and what's at stake in the introduction, starting with ACI and how it is measured then how it is modeled and how the two disagree. I understand how difficult it is to compress many of these complicated concepts such as precipitation suppression or enhanced entrainment-induced evaporation which introduce uncertainty, but this paper would benefit from more explanation of these processes.

We sincerely appreciate your thoughtful and constructive feedback. All comments have been carefully considered, and the manuscript has been revised accordingly.

For the data and method and section, I think this paper would benefit from discussion of the satellite products chosen. The literature suggests that most satellites with same measurement types are in good agreement, but I'm curious about the omission of MODIS Terra as it would provide more temporal resolution. Figures 2 and 4 are great for showing that models qualitatively match Nd-LWP and aerosol extinction coefficient-Nd relationships in observations, but are more

exaggerated. The comparisons between observations and model parameters is well summarized at the end of each section.

We restricted the satellite data set to the curated CALIOP–MODIS Aqua collocation introduced in Painemal et al. (2020) and Li et al. (2025). That product was designed around daytime MODIS Aqua cloud retrievals processed with the CERES Ed. 4.0 algorithms and paired with CALIOP-S aerosol extinction, providing consistent sampling and reduced retrieval artifacts.

Adding MODIS Terra would increase temporal sampling, but its morning overpassing time (~10:30 LT) does not coincide with the early-afternoon (~13:30 LT) overpassing time of CALIOP on CALIPSO. Hence our collocation specifically targets the afternoon local overpass to keep aerosol-cloud relationships at a single time of day. Combining Terra with Aqua would therefore introduce diurnal aliasing into the ACI relationships derived. Prior work shows that the aerosol indirect effect and related cloud adjustments vary over the diurnal cycle (Diamond et al., 2020; Smalley et al., 2024), so fixing the local time is critical for a clean evaluation.

Finally, the Painemal et al. (2020) collocation/aggregation strategy was built and quality-controlled for Aqua, and it demonstrably reduces  $N_d$  bias, replicating that framework for Terra would require additional harmonization. We are interested in building such morning database for future work.

**We have added the associated discussion in the revised Section 2.1:**

'Moreover, the aggregated collocation method significantly reduces the MODIS Aqua  $N_d$  bias (Painemal et al., 2020), resulting in a relationship between aerosol and cloud properties less affected by artifacts. Note that to avoid diurnal variations in aerosol-cloud relationships, we fix the sampling to the Aqua local-afternoon overpass and do not merge with Terra morning orbits, while extending the collocation and quality-control framework to Terra is left for future work.'

While very thorough, I can't help but feel as though the section 3's focus on seasonal comparisons between satellite and model parameters and relationships is unnecessary to this body of work,

especially with the end of that section mentioning its limitations and the need to separate by regime to disentangle meteorological variability which I think is the meat of this body of work.

We appreciate the suggestion, we have now significantly streamline Section 3 to make the discussion concise, and compelling for the necessitate of meteorological-regime-based analysis of the ACI in Section 4.

The comparisons between observations and the E3SMv2 models in each regime are thorough and any discrepancies have a one or more hypothesis supported by the literature. The paper concludes with many suggestions of next steps on how to improve model representation of observations, leaning on the difficulties of overcoming satellite measurement uncertainty. I think it would be useful to sort of rank future changes that are most feasible or important to implement. I'm excited to see future work in other regions, especially of the number of regimes found and how these relationships change, especially with more drastic differences in sources of aerosol.

We appreciate the suggestion, and Review 1 also raised similar concerns. We have added the following discussion in the conclusion section:

'In terms of the feasibility of potential model improvements, we think that a feasible approach would be the fine-tuning of the microphysical parameterization, ideally constrained by high-resolution observational data from field campaigns such as ARM. This may reduce the persistent uncertainties in simulating aerosol-cloud interactions, particularly under the dynamic meteorological transitions typical of the ENA region. Furthermore, emulation from high-resolution modeling (e.g., LES) of cloud and rain microphysics processes can be used to replace the bulk microphysics scheme, which can contribute to better performance with manageable cost as shown in previous studies such as Gettelman et al. (2021). Increasing spatial resolution is also feasible in a regionally refined mesh, and increasing vertical resolution might follow, but both would noticeably increase computational cost, so trade-offs should be considered with caution. Lastly, the development of new schemes that bridge the gap between shallow and deep cloud regimes remains particularly challenging, as current large-scale model schemes still treat them separately.'

**Specific comments:**

L48 Nitpicky, but satellite remote sensing's main advantage over in-situ cloud measurements is the spatial extent. The temporal resolution is often worse, especially of polar orbiting satellites.

We have changed this statement to 'Satellite remote sensing observations are essential in efforts to quantify the cloud adjustment to aerosol perturbations, by providing spatially extensive datasets.'

L49 It would be good to explicitly cite more of the numerous papers advancing ACI using satellite data.

Following your suggestion, more papers on the satellite advance on the ACI are cited:

'Numerous studies using satellite data have demonstrated a significant relationship and progressively advanced our understanding of cloud adjustments to aerosols (Bellouin et al., 2020; Diamond et al., 2020; Yuan et al., 2023; Feingold, et al., 2025; Goren et al., 2025).'

L67 It would help to introduce the inverted V-shaped relationship observed in satellite retrievals in the previous paragraph. It is currently explained in detail much later in L283.

Thanks for the suggestion, the inverted V-shaped relationship is now introduced in detail in the introduction, rather than Section 3.

L159 It would help to include a typical range of relative error values in retrieval and r-values of comparisons if available to better quantify what is meant by significant and decent.

We have now included the error estimate from previous study as below:

'...previous studies have shown that the  $N_d$  compares well with measurements from 11 aircraft campaigns, demonstrating a decent correlation when sampling the marine stratocumulus clouds, with  $r^2$  values of 0.5~0.8 (Gryspeerdt et al., 2022). Therefore, to minimize known retrieval uncertainties, we focus on low-level liquid clouds where satellite  $N_d$  shows the strongest aircraft

agreement and typical normalized root mean squared deviation of ~30-50 % (Gryspeerdt et al., 2022).'

**L242, I think figure 1 would benefit from a 5th column of annual statistics.**

**The annual statistics are now added as 5th column in Figure 1:**

**Figure 1.** Violin plots of cloud droplet number concentration ( $N_d$ , top panels) and cloud liquid water path (LWP, bottom panels) from satellite retrievals (purple) and E3SMv2 simulations (blue), during winter, DJF (a, f), spring, MAM (b, g), summer, JJA (c, h), fall, SON (d, i), and Annual (e, j). The mean value is indicated by the color-coded dot. The smoothed shape of each violin shows the Gaussian kernel density estimate (KDE). From top to bottom within each violin, the box plot lines represent the third quartile (Q3, 75th percentile), median (Q2, 50th percentile), and first quartile (Q1, 25th percentile), respectively. The upper whisker extends to Q3 + 1.5 × IQR (interquartile range), and the lower whisker extends to Q1 – 1.5 × IQR.

And the corresponding description has been revised to:

'Figure 1 illustrates the seasonal variations in the Nd and LWP for low-level clouds over the ENA, from satellite (MODIS) retrievals and E3SMv2. Annual means are  $88.33 \pm 91.67$  cm-3 and  $82.17 \pm 68.61$  g m-2 for satellite  $N_d$  and LWP, and  $65.84 \pm 38.59$  cm-3 and  $77.49 \pm 73.41$  g m-2 for E3SMv2 (Fig. 1e and 1j).'

L288 It would be helpful to provide ranges of Nd of these regimes found in previous work for comparison.

Thanks for the suggestion. We have included the previous reported  $N_d$  ranges as below:

'In the satellite observations (Fig. 2a), increasing  $N_d$  suppresses precipitation, leading to LWP accumulation at low  $N_d$ . At higher  $N_d$ , the LWP response turns negative, consistent with enhanced entrainment and evaporative losses. These results are broadly consistent with prior satellite studies over marine stratocumulus: an inverted-V LWP- $N_d$  relationship has been reported for  $N_d$  ranges of  $\sim$ 10-300 cm-3 in the southeast Pacific (Goren et al., 2025), globally (Gryspeerdt et al., 2019; 2022), and  $\sim$ 7-400 cm-3 for subtropical stratocumulus (Possner et al., 2020).'

L294 Missing space after Nd.

Thanks, it is corrected.

L332 The explanation of enhanced sea salt emissions with increased wind speed makes sense and should be observed globally, but is the seasonal dependency on the under and overestimation of the model aerosol extinction coefficient regionally dependent then if dust is a factor? Additionally, it would help to provide more context of where the transported dust in the ENA comes from.

The seasonal dependence of E3SM's  $\sigma_{EXT}$  bias in the ENA is consistent with broader, global tendencies in the model's dust cycle rather than a signal unique to ENA. In E3SM, dust vertical/long-range transport is generally underrepresented relative to CALIPSO (Feng et al., 2022), and regional dust AOD differences persist even when the global mean is constrained. Thus,

the ENA seasonal pattern likely reflects these global dust process biases expressed through regional meteorology and sampling.

Overall, ENA dust originates from North Africa (Sahara Desert) and arrives episodically, the winter-spring synoptic events can drive northwestward intrusions that sporadically impact the ENA, and hence potentially impacts the seasonality on the aerosol extinctions (Logan et al., 2014; Rodríguez and López-Darias, 2024).

**We have revised the associated discussion in Section 3.3 of the manuscript:**

'The observed high  $\sigma_{EXT}$  in cold seasons reflects coarse-mode contributions from both enhanced sea-salt emissions under strong MBL winds and episodic Saharan dust intrusions that reach the ENA via the synoptic northwestward transport (Logan et al., 2014; Gläser et al., 2015; Rodríguez and López-Darias, 2024). E3SMv2 likely underpredicts this signal due to low sea spray (Burrows et al., 2020) and an underrepresentation of dust vertical extent and transport (Feng et al., 2022), a broader model tendency that appears over the North Atlantic as well (H. Wang et al., 2020; Qin et al., 2024).'

L365 It would be helpful to provide the sensitivity values from the cited studies for comparison.

We have added the following discussion:

'The positive  $ACI_N$  reflects the Twomey effect and lies within reported satellite ranges over marine stratocumulus and the eastern Atlantic. For example, McCoy et al. (2017) found a log-log slope of 0.31 between  $N_d$  and sulfate mass, Jia et al. 2021 reported 0.14-0.51 for  $N_d$  versus AOD over oceans, and recent reviews summarize satellite-based susceptibilities of about 0.1–0.7 depending on sampling and aerosol proxies (Gryspeerdt et al., 2023).'

L513 It would be helpful to provide a range of the peak of the inverted V-shape across the regimes in text to highlight this point.

We have added the following quantitative range of peaks:

'Across regimes, the peak in the satellite data of the inverted LWP- $N_d$  curve occurs at  $N_d \approx 15$ -96 cm-3 (particularly, 96.5 cm-3 at Regime 4), and at LWP  $\approx 101$ -136 g m-2; whereas the E3SMv2 peaks at a much narrower  $N_d \approx 15$ -19 cm-3 with higher peak LWP  $\approx 142$ -171 g m-2.'

L523 While I understand that £0 is bulk, wouldn't separating by the peak of the inverted V-shape illustrate your points more of the rapid decrease?

We have computed the below table of the post-peak ( $N_d >$  peak binned  $N_d$ ) log-log slopes for LWP vs  $N_d$  in each regime.

| Regime          | Peak LWP          | Peak LWP-N d | Slope ± SE from    | Slope ± SE from    |
|-----------------|-------------------|-------------------------|--------------------|--------------------|
|                 | corresponding     | binned mean $N_d$       | Sample > peak      | Sample > peak      |
|                 | binned $N_d$      | (E3SM)                  | bin (satellite)    | bin (E3SM)         |
|                 | (satellite)       |                         |                    |                    |
| 1 (Pre-Trough)  | 15.42             | 18.90                   | $-0.179 \pm 0.022$ | $-0.377 \pm 0.017$ |
| 2 (Post-Trough) | 18.90             | 15.42                   | $-0.214 \pm 0.020$ | $-0.390 \pm 0.017$ |
| 3 (Ridge)       | 18.90             | 15.42                   | $-0.176 \pm 0.008$ | $-0.327 \pm 0.008$ |
| 4 (Trough)      | 96.49 (last $N_d$ | 15.42                   | N/A due to too     | $-0.417 \pm 0.042$ |
|                 | bin)              |                         | few samples        |                    |

In all four regimes, the post-peak slopes are negative for both datasets, and E3SM is consistently more negative (steeper decline) than satellite. This preserves the same sign and ordering as the bulk analysis, so the qualitative message is unchanged. Note that in regime 4 satellite has very few valid points and a large standard error since the LWP peak at the last  $N_d$  bin, which would invite distracting caveats if moved into the main text.

Therefore, as  $\mathcal{L}_0$  is bulk metric by design. Mixing it with a subset-conditioned (post-peak) slope in the main text can blur the narrative and burden readers with extra conditioning details.

Figure 8 Nitpicky, but I think this figure would benefit from labels of the tickmarks between 10 and 50 as a significant part of the data is within that range.

The Figure 7 & 8 are replotted with more information on the x-axis tickmarks.